# Computational geometry analysis of dendritic spines by structured illumination microscopy

Yutaro Kashiwagi[1], Takahito Higashi[1], Kazuki Obashi[1], Yuka Sato[1], Noboru H. Komiyama[2], Seth G.N. Grant [2] & Shigeo Okabe [1]

Dendritic spines are the postsynaptic sites that receive most of the excitatory synaptic inputs, and thus provide the structural basis for synaptic function. Here, we describe an accurate method for measurement and analysis of spine morphology based on structured illumination microscopy (SIM) and computational geometry in cultured neurons. Surface mesh data converted from SIM images were comparable to data reconstructed from electron microscopic images. Dimensional reduction and machine learning applied to large data sets enabled identification of spine phenotypes caused by genetic mutations in key signal transduction molecules. This method, combined with time-lapse live imaging and glutamate uncaging, could detect plasticity-related changes in spine head curvature. The results suggested that the concave surfaces of spines are important for the long-term structural stabilization of spines by synaptic adhesion molecules.

[1] Department of Cellular Neurobiology, Graduate School of Medicine, the University of Tokyo, Tokyo 1130033, Japan. [2] Genes to Cognition Program, Centre for Clinical Brain Sciences, University of Edinburgh, Edinburgh EH16 4SB, UK. Correspondence and requests for materials should be addressed to S.O. (email: okabe@m.u-tokyo.ac.jp)

Dendritic spines are submicron-scale structures protruding from neuronal dendrites that receive excitatory synaptic inputs from afferent axons[1,2]. Precise measurement of spine morphology and objective analysis of large numbers of spines are required to understand both the physiological regulation of synaptic transmission and pathological changes in this process[3]. Dendritic spines exhibit a high degree of structural variability: their sizes vary over more than one order of magnitude[4], and their shapes range from thin elongated filopodia-like protrusions to round mushroom-like structures[5]. Recent analyses of spine images obtained by quantitative and super-resolution optical imaging argued against the conventional categories of thin, mushroom, or stubby spines, and suggested that continuous morphological variables should be used instead[6,7]. Structural studies of spines by electron microscopic (EM) reconstruction also opposed the idea of well-defined categories of dendritic spines[3]. In light of these developments, it is necessary to develop new strategies that allow comprehensive analysis of highly variable spine structures.

Traditionally, structural analysis of dendritic spines has relied on reconstruction of EM images[3]. Although new technologies for automated acquisition of serial EM images greatly accelerated quantitative analysis of spine morphological features[8,9], the lack of information about the dynamic properties of spines limits the application of these methods to functional studies. Recent progress in multiple technologies related to super-resolution imaging has raised the possibility of accurate and high-throughput imaging of submicron spine structures[7,10,11]. Several recent studies proposed computational methods for objective shape classification and modeling of dynamic spine behavior. First, spine shape analysis based on a commercially available software package, such as Imaris software, was proposed[12]. This study utilized three structural parameters and simple formulas for spine classification. Second, Rodriguez et al.[13] proposed a spine analysis algorithm in the platform of a feely distributed software NeuronStudio. With this tool, spine classification is performed by a decision tree with three key parameters, aspect ratio (the extent of shape elongation), head-to-neck ratio (ratio of their diameters), and spine head diameter. Small numbers of parameters for spine shape in these studies may not be adequate for complex three-dimensional (3D) spine images obtained by super-resolution microscopy. Third, spine classification by semi-supervised learning in combination with multiple spine structural parameters was reported[14]. This method has an advantage of requiring a relatively small number of a training dataset, and seven features were reported to be effective for spine classification. Because the details of the datasets and the selection of effective features were not provided, general applicability of this method to other image samples should be judged by further trials. In summary, these efforts have not yet fully utilized the high 3D resolution of new imaging modalities capable of overcoming the diffraction barrier.

Here, we describe an accurate method for spine morphological measurement based on structured illumination microscopy (SIM) and subsequent data conversion to the surface mesh data. Dimensional reduction and supervised machine learning enabled us to perform objective spine classification and identify morphological impairments caused by genetic mutations in key signal transduction molecules. Time-lapse SIM imaging could be combined with this analytical method to generate shape transition diagrams. Furthermore, analysis of spines exposed to locally uncaged glutamate revealed stable concave surfaces on activated spine heads, which may serve as core structural elements of synaptic plasticity.

## Results and Discussion

### A method for measuring nanoscale surface geometry of spines.

Our analytical pipeline for spine computational geometry consists of three steps (Fig. 1a, b). First, three-dimensional SIM (3D-SIM) images of dendritic segments in dissociated hippocampal neurons labeled with either fluorescent proteins or the lipophilic dye DiI were obtained, and voxel data of fluorescent spines were converted to the surface mesh data using automated algorithms. Second, the numerical features of 3D spine morphology were calculated by computational geometry from a dataset for a large population of spines. Finally, the high-dimensional data of recorded numerical features (descriptors) were transformed to a space with fewer dimensions by principal component analysis (PCA), followed by support vector machine (SVM)-based shape classification. To validate this method, we analyzed neurons derived from heterozygous synGAP mutant mice (synGAP$^{+/-}$)[15] or mice harboring a knock-in of a kinase-dead allele of Ca$^{2+}$/calmodulin-dependent protein kinase IIα (CaMKIIα$^{K42R/K42R}$)[16] and confirmed that this method can detect mutated gene-specific spine structural changes.

We obtained 3D-SIM images of cultured hippocampal neurons expressing GFP (Fig. 1a). 3D-SIM microscopy theoretically offers a twofold resolution gain in both the lateral and axial directions in comparison with conventional wide-field microscopy[17]. This property was confirmed by measuring intensity profiles of diffraction-limited images and SIM images of fluorescent beads (Supplementary Fig. 1). We found that correction of spherical aberration by precise matching of refractive index and stabilization of sample temperature improved the resolution close to the theoretical limit and enabled reliable detection of dendritic spines protruding vertically in the axial direction (Supplementary Fig. 1). We next tested multiple image segmentation algorithms and found that Otsu's method, modified for multi-level thresholding, combined with the technique of geodesic active contours[18], was resistant to variations in dendritic morphology and image intensity (Supplementary Fig. 2). Individual spines were detected and isolated automatically using a custom software (Fig. 1a, Supplementary Fig. 3). The method is based on fitting of dendritic shafts with elliptic cylinders followed by detection of structures outside of the cylinder as spine candidates (Supplementary Fig. 4). These spine candidates were further sorted automatically by criteria based on their volumes and shapes (see the Methods section for the details). This method is based on the morphological criteria and cannot distinguish very short spines from small raised structures on dendritic shafts. Nevertheless, the detection criteria of spines are objective and the results are reproducible. After spine isolation, polygonal meshes of isosurfaces were extracted from the spine voxel data using the marching cube algorithm[19] (Fig. 1b, Supplementary Fig. 3). These spine mesh objects were automatically analyzed by techniques for feature extraction from 3D mesh models[20] or by discrete differential-geometry operators[21] (Supplementary Fig. 5). Ultimately, we obtained 10 descriptors of spine shape from experimental datasets of more than a 1000 spines. These descriptors include both basic shape features (e.g., length, surface area, and volume) and more complex shape and surface features (e.g., convex hull, mean curvature, and Gaussian curvature).

### Surface geometry of spines and quantitative analysis.

To confirm the accuracy of multiple shape descriptors extracted from SIM imaging, we performed quantitative comparison of SIM-based data from fixed samples with surface mesh data generated by EM reconstruction of identical dendritic segments (Fig. 1c). All spines detected in EM images were also recovered in SIM-based surface mesh data irrespective of the protrusion direction. A minor fraction of spines (<15%) overlapped with adjacent spines and were excluded from the analysis. Mean curvature plots revealed that negative and positive curvatures of spine surfaces

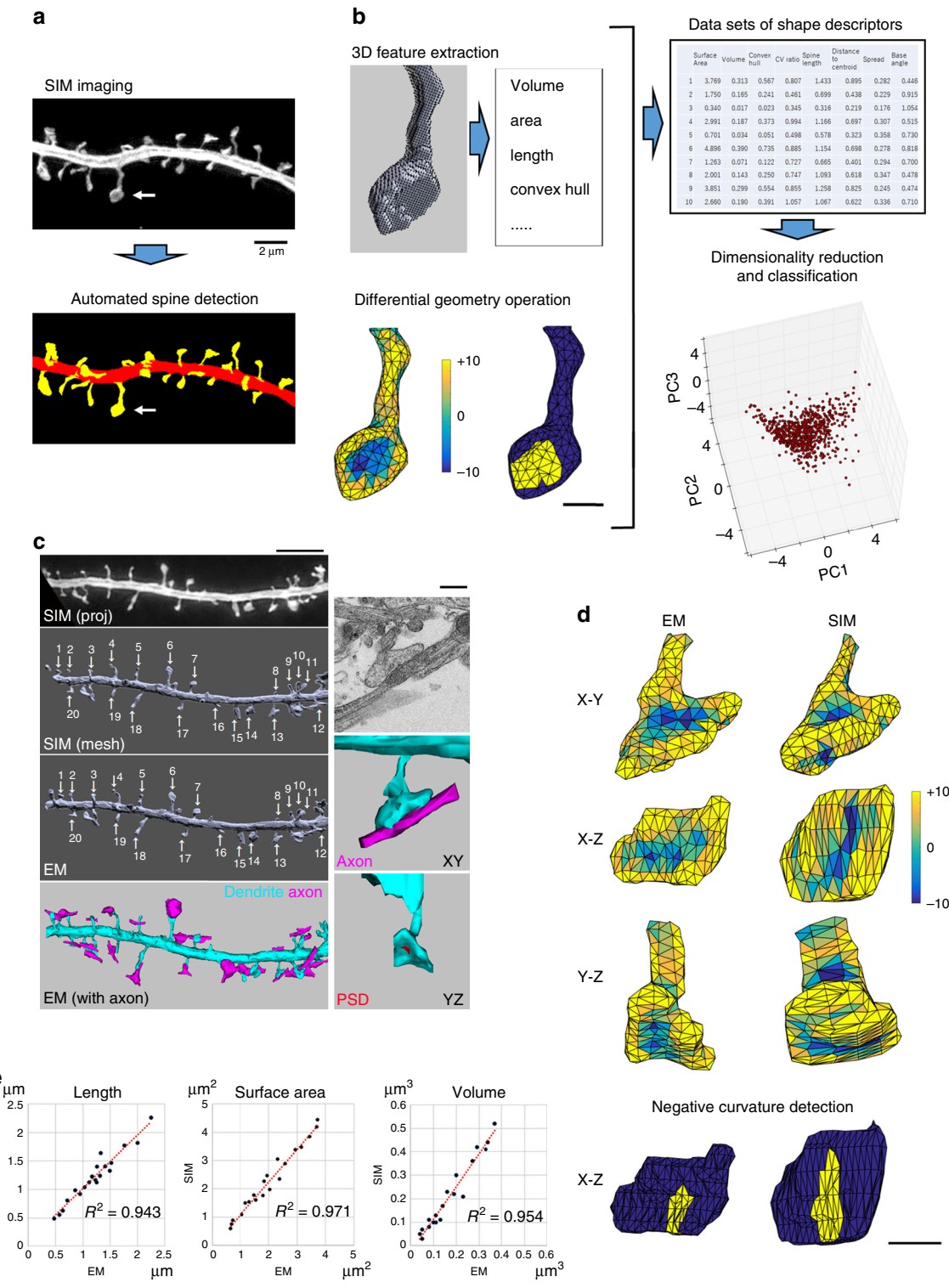

were preserved between the SIM-based and EM-based 3D data (Fig. 1d). The concave surface was present in 61% of spines of which volumes were more than 0.18 μm³ (this spine population corresponds to 25% of the total spines and 84% of them are classified as mushroom spines using our machine-learning method. See the next section for the detection of mushroom spines.) and matched the position of the synaptic junction and postsynaptic density (PSD) (Fig. 1c, d, Supplementary Figs. 6 and 7). Thus, the spine concave surface may be a biologically

important feature that reflects the presence of junctional complex between pre- and postsynaptic membranes. The values of descriptors for basic shape features (length, surface area, and volume) were highly correlated between surface mesh data extracted from 3D-SIM and EM images (Fig. 1e), indicating the precision of SIM-based spine analysis.

Lengths of spines protruding from dendritic shafts at different angles were measured in both 3D-SIM and EM images (Supplementary Fig. 8). Both horizontally and vertically

**Fig. 1** A method for measuring the surface geometry of dendritic spines. **a** Acquisition of 3D-SIM image of dendrites and automatic detection of dendritic spines. Arrows indicate the same dendritic spine shown in panel (**b**). Bar: 2 μm. **b** Process of spine geometry analysis. Individual spine mesh objects can be visualized as pseudocolor or shaded surface images (mesh feature extraction). Differential geometry can be calculated, and the parameters can be mapped onto the surface (differential-geometry operation). Bar: 500 nm. After calculation of multiple shape descriptors, the datasets are further analyzed by the techniques of dimensionality reduction and automatic classification using machine learning. **c** Comparison of 3D-SIM images and reconstruction of EM images from the identical dendritic segment. Left side column shows a lower-magnification view of a dendritic segment in a SIM projection image [SIM (proj)], a reconstructed surface view of SIM and EM data [SIM (mesh) and EM], and a reconstructed view of the dendrite with presynaptic components [EM (with axon)]. Numbers (1–20) indicate the corresponding spines. Right column shows a higher-magnification view of spine 13, with raw single plane EM (upper), reconstructed EM data with axon (middle), and reconstructed EM data with PSD (lower). Bars: 5 μm for left column, 500 nm for right column. **d** Comparison of spine no. 13 in panel (**c**), reconstructed from EM and SIM data. Surface mean curvature is shown by pseudocolor mapping. The lower image pair shows the areas with the smallest negative value of mean curvature in yellow. Bar: 500 nm. **e** Relationship of basic shape parameters (length, surface area, and volume) calculated from EM and SIM data. High values of coefficient of determination indicate the possibility of estimating the absolute shape parameters from SIM images after appropriate conversion ($n = 20$ spines in a single reconstructed volume)

---

protruding spines could be measured with similar accuracy in their lengths. Although spine neck width is an important parameter related to spine functions, we did not include spine neck width as a spine shape descriptor from the following reasons. First, PCA and SVM-based classification requires parameters that can be measured in all spines, but spine necks do not exist or difficult to define in thin or stubby spines. Second, spine neck widths in the axial direction were overestimated due to lower axial resolution of SIM (Supplementary Fig. 8d). This resolution asymmetry complicates interpretation of the data from spines protruding in different directions.

We next investigated whether the multiple descriptors of spine shape features would be useful for representing a dataset for a large spine population with continuous morphological variables. From our initial analysis of independence among descriptors in a dataset of 1335 spines, we selected five descriptors based on two criteria. First, we selected two descriptors that reflect principal structural features (length and volume). Second, three other descriptors that showed high independence (the averages of pairwise correlation coefficients were <0.3) were selected. PCA was performed with these five descriptors to obtain spine distribution in the new feature space (Fig. 2a). The first three features (principal components 1–3, PC1–3) covered about 93% of the variance in the data. Principal components are normalized linear combinations of the original descriptors and reflect specific morphological properties of spines. Spine distribution in Fig. 2a, b indicates that spines with different sizes are aligned along the axis of PC1 [No. 1 (large) and No. 10 (small)] and spines with different length–width ratio are distributed along the axis of PC2 [No.6 (thin) and No. 8 (thick)]. Along the axis of PC3, spines with different head sizes relative to total spine volumes are aligned [No. 3 (prominent spine head) and No. 7 (without spine head)]. In summary, PCA was effective in extracting three independent structural features of dendritic spines.

**Spine structural features with impaired synaptic plasticity.** Spine distribution in the feature space exhibited a continuum of morphologies, supporting the idea that the conventional categorization into thin, mushroom, or stubby spines does not reflect the presence of discrete subclasses. On the other hand, we noticed that prototypical examples of mushroom spines (spines nos. 1–4 in Fig. 2b) were located closely to each other in the feature space, raising the possibility of automatically identifying mushroom spines by supervised machine learning. Using the manually classified dataset, we trained a SVM classifier with a nonlinear kernel. After hyperparameter tuning, the accuracy of the classifier was 88.4%, which is comparable with or better than that of other spine classification methods[13,22] or manual classification by multiple operators (Fig. 2c and Supplementary Table 1). This method of

spine morphological analysis could be applied to neurons stained with the lipophilic dye DiI after fixation (Supplementary Figs. 9 and 10). Feature selection and dimensional reduction by PCA can improve the performance of classifiers by removing redundant or irrelevant features. To test the contribution of preprocessing the data by PCA, we compared the performance of SVM classifiers with or without PCA. When GFP-labeled spine images were used for both training and test, the two methods showed similar performance (accuracy = 90.3% without PCA, accuracy = 89.3% with PCA), but the method without PCA showed slight decline in performance with the test data from DiI-labeled spines after training with the data from GFP-positive spines (accuracy = 83.0% without PCA, accuracy = 86.4% with PCA), suggesting that PCA may be effective in eliminating parameters irrelevant to structural characteristics of dendritic spines.

We next applied this automated classification method to the detection of spine structural changes associated with mutations in genes encoding plasticity-related signaling molecules. To this end, we obtained 3D-SIM images of neurons derived from heterozygous synGAP mutant mice (synGAP$^{+/-}$)[15] or mice harboring a knock-in of a kinase-dead allele of Ca$^{2+}$/calmodulin-dependent protein kinase IIα (CaMKIIα$^{K42R/K42R}$)[16] (Fig. 2d). Both mutations result in severe impairment of long-term potentiation in the hippocampus[15,16]. To avoid possible bias of GFP transfection into specific types of neurons, we stained randomly selected neurons by applying DiI (Supplementary Figs. 9 and 10). SIM-based quantitative analysis revealed changes in spine morphology specific for each mutation (Fig. 2e). Neurons from either synGAP$^{+/-}$ mice or CaMKIIα$^{K42R/K42R}$ mice formed mushroom spines with reduced volume, with no change in the volume of non-mushroom spines (Fig. 2f). The two mutants exhibited distinct spine length phenotypes, with shorter mushroom spines in synGAP$^{+/-}$ neurons and longer non-mushroom spines in CaMKIIα$^{K42R/K42R}$ neurons. The results suggest a more specific impairment in mushroom spines with the synGAP mutation, and a specific role of CaMKIIα in suppression of long non-mushroom spines. These results demonstrate the potential of this analytical system to detect spine morphological changes associated with dysfunction of specific signaling pathways.

**Computational geometry of spines in vivo.** Spine geometrical analysis requires high-resolution imaging of dendritic spines. SIM imaging of dissociated neurons in culture has sufficient resolution, but alternative approaches are required for the analysis in intact brain tissue. To test if confocal laser scanning microscopy is suitable for spine geometrical analysis, we performed in silico analysis of spine shape degradation by optical blur (Supplementary Fig. 11a–c). By narrowing confocal aperture [0.5 airy unit (AU)], the resolution was sufficient to detect spine head curvature

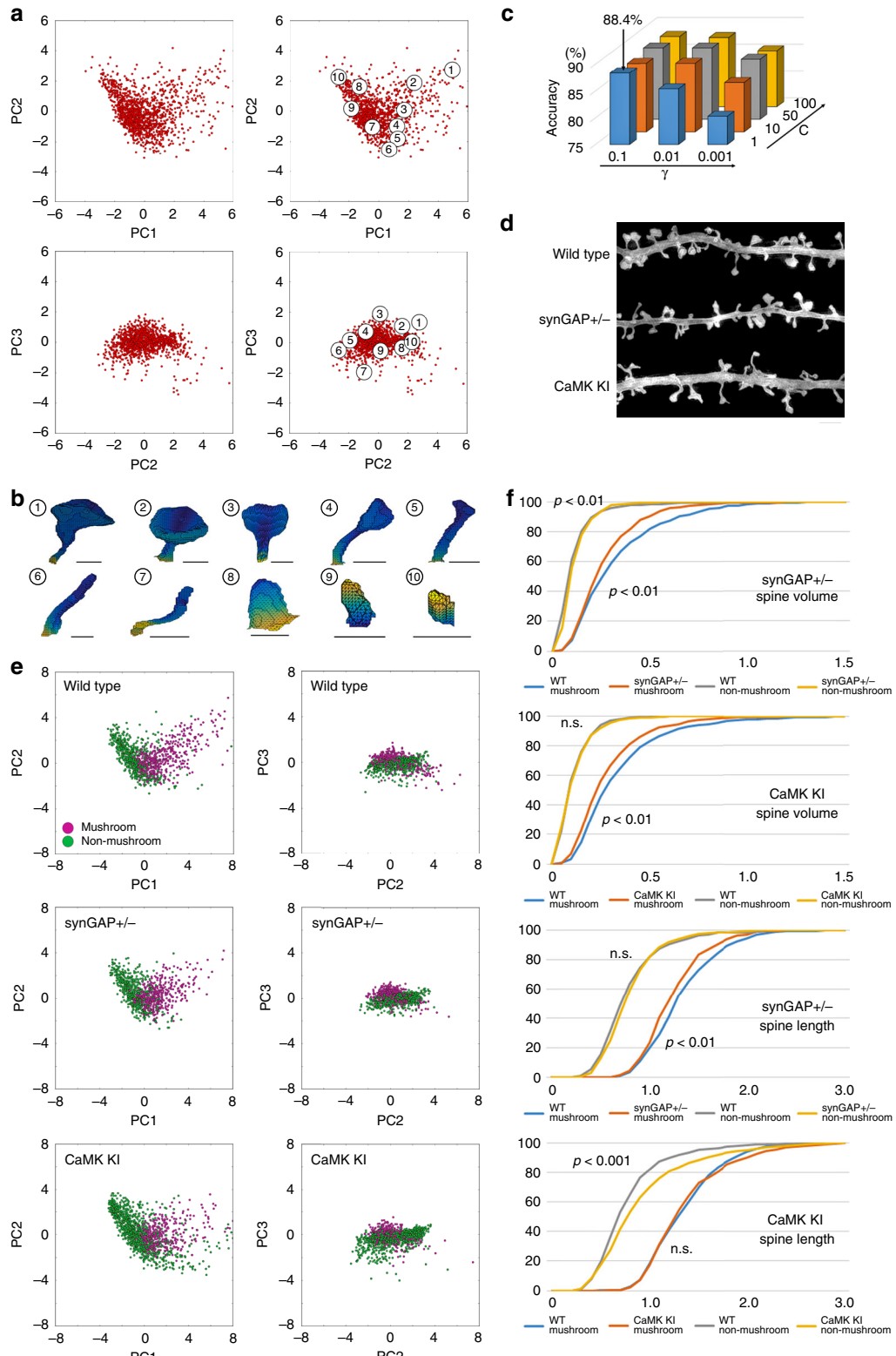

for large spines (>0.18 μm$^3$). Because SIM analysis of cultured neurons showed that only 13% of middle-to-small-sized spines (<0.18 μm$^3$) had concave surfaces, the confocal scanning microscopy may still be useful in feature extraction of large spines in intact tissue. We also confirmed that spine size distribution was similar between hippocampal neurons in culture and in the intact hippocampal tissue[23] (Supplementary Fig. 12). Therefore, we

expected that geometrical features of large spines could be detected by high-resolution confocal microscopy applied to intact brain tissue.

We applied confocal microscopy with confocal aperture of 0.5 AU to CA1 pyramidal neurons expressing YFP in fixed brain sections (Supplementary Fig. 11d, f). Horizontally and vertically protruding spines could be detected and converted to the surface

**Fig. 2** Spine shape analysis based on dimensionality reduction and SVM-based labeling of large spine datasets. **a** Distribution of spines in the feature space with axes corresponding to PC1, PC2, and PC3. The right two graphs show the positions of spine examples in panel (**b**) ($n = 1335$ spines from 86 cells in four independent culture preparations). **b** Examples of spines in the feature space shown in panel (**a**). Bar: 500 nm. **c** Hyperparameter tuning for the SVM classifier. Accuracy was highest (88.4%) with values of $C = 1$ and $\gamma = 0.1$. **d** SIM projection images of wild-type, synGAP$^{+/-}$, and CaMKII$\alpha^{K42R/K42R}$ neurons. Bar: 2 μm. **e** Distribution of spines with different genotypes mapped into the feature space and classified as mushroom (magenta dots) or non-mushroom (green dots) spines by SVM ($n = 1095$ spines from 43 neurons in three independent culture preparations for the wild-type, 1015 spines from 44 neurons in three independent culture preparations for synGAP$^{+/-}$, and 1392 spines from 48 neurons in three independent culture preparations for CaMKII$\alpha^{K42R/K42R}$). **f** Cumulative distribution curves of spine volume and length with different genotypes after classification into mushroom and non-mushroom spines (Kolmogorov–Smirnov test; synGAP experiments: $n = 487$ for wild-type mushroom spines, $n = 608$ for wild-type non-mushroom spines, $n = 466$ for synGAP$^{+/-}$ mushroom spines, $n = 549$ for synGAP$^{+/-}$ non-mushroom spines; CaMKII experiments: $n = 549$ for wild-type mushroom spines, $n = 698$ for wild-type non-mushroom spines, $n = 424$ for CaMKII$\alpha^{K42R/K42R}$ mushroom spines, $n = 968$ for CaMKII$\alpha^{K42R/K42R}$ non-mushroom spines)

mesh data. Consistent with the prediction of in silico analysis, concave surfaces were detected in large spines (~20% of the total spine population). We conclude that this application in intact tissue is useful in geometrical analysis of spine head surface, which reflect the presence of the junctional complex between pre- and postsynaptic membranes (Supplementary Figs. 6 and 7)

Spine population data obtained by confocal microscopy of intact tissue may be useful in analysis of spine phenotypes based on dimensional reduction and machine learning. To test this possibility, we collected the data of spine surface geometry in tissue sections ($n = 165$) and compared the distribution in the feature space with the data from cultured neurons (Supplementary Fig. 13a). Distributions of spine geometrical features from samples in culture and in vivo show high similarity, and the positions of typical mushroom, thin, and small spines are preserved. Spine size distribution measured from the confocal imaging of intact tissue [$0.081 \pm 0.091$ μm³ (mean ± SD, $n = 165$)] was comparable with that from both cultured hippocampal neurons [$0.079 \pm 0.078$ μm³ (mean ± SD, $n = 1335$)] and EM reconstruction of in vivo spines [$0.076 \pm 0.082$ μm³ (mean ± SD, $n = 938$)][23]. SVM-based classifier trained with the dataset obtained from GFP-expressing neurons in culture was applied to the in vivo dataset to identify mushroom spines (Supplementary Fig. 13b). The accuracy of the classifier was 88.0%, indicating slight decline in performance in comparison with the data obtained from cultured neurons by SIM (89.3% accuracy). In summary, the methods of computational geometry, dimensional reduction, and SVM-based shape classification are useful for analyzing high-resolution confocal images of dendritic spines in intact tissue.

**Shape transition of spines studied by time-lapse 3D-SIM.** An advantage of SIM-based geometrical analysis is its potential use in the studies of spine dynamics and activity-dependent regulation. To follow temporal changes in spine shape, we performed time-lapse 3D-SIM imaging of living hippocampal neurons expressing GFP (Supplementary Fig. 14a, b). Comparison of live and fixed spines confirmed that the reconstructed mesh structures from live and fixed cells were of comparable quality, and the concave surface in the spine head was preserved (Supplementary Fig. 14c). Using PCA, we mapped the trajectories of shape transitions of individual spines in the feature space (Fig. 3a) and generated a 3D map depicting the behavior of the spine population (Fig. 3b). We found that spines in different domains of the feature space behaved differently. For example, the large mushroom-shaped spine in Fig. 3a (magenta arrows) moved bidirectionally in the upper right and lower left directions. The medium-sized spine (orange arrows) exhibited short trajectories, resembling a random walk process, whereas the trajectory of the small spine (green arrows) was in the upper left direction. To further clarify the overall tendency of the shape transition, we generated a diagram

in the feature space that shows the direction and length of trajectories partitioned into voxels with edge length of $1 \times$ standard deviation (SD) (Fig. 3c). This diagram further confirmed the relationship between spine shape transition and spine shape features. Using the SVM classifier combined with trajectory analysis, it was possible to label spines with groups 1–3 (Supplementary Fig. 15). Spines of group 1 (small mushroom spines without an orientation preference in their trajectories), group 2 (large mushroom spines with preferred trajectories along the axis of medium/thin and large/round shape features), and group 3 (non-mushroom spines) had different shape characteristics and dynamics (Fig. 3d).

The three groups of spines overlapped in the feature space, and their distribution did not reflect the existence of distinct shape classes. However, spines may have additional geometric features that would aid in understanding the mechanisms underlying the maintenance of their shapes and behaviors. Correlative analysis of 3D-SIM and EM images indicated that the concave surfaces of spine heads tended to be associated with the presynaptic component and formed synaptic junctions (Fig. 1c, d). When the concave surfaces of spine heads were mapped at multiple time points, their stability differed markedly among groups (Fig. 3e). Most group 2 spines maintained the concave surface during imaging sessions of 60 min (10 out of 11 spine time-lapse images), whereas group 1 spines changed their shape and either lost the concave surface or changed its position (5 out of 7 time frames, 3 spines, Fig. 3f). Group 3 spines formed the concave surface less frequently than those of the other two groups, and exhibited a strong tendency to progressively decrease their size, suggesting that this spine population was undergoing the process of retraction. This result indicates that formation of stable junctions between presynaptic and postsynaptic compartments is associated with maintenance of spine shape and dynamics.

**Changes in spine geometry by synaptic plasticity.** In the feature space, group 2 spines exhibited the longest trajectories along the axis of PC1, indicating that this group undergoes larger changes in size, despite the stability of their synaptic junctional membranes. This observation is consistent with multiplicative dynamics, a model based on in vivo two-photon imaging, in which the magnitude of changes in spine size is proportional to the size of the spine[4]. In the feature space, group 2 spines also overlapped with spines classified as mushroom spines by the SVM classifier, which were decreased in size by mutations in genes related to synaptic plasticity (Fig. 2e, f). We therefore investigated whether induction of structural plasticity regulates the surface geometry of spines, especially concave surfaces of synaptic junctions, and contributes to subsequent maintenance of the enlarged spine structure. Local uncaging of glutamate in $Mg^{2+}$-free solution induced a large transient increase in spine volume, followed by a stable plateau lasting more than 60 min (Fig. 4a).

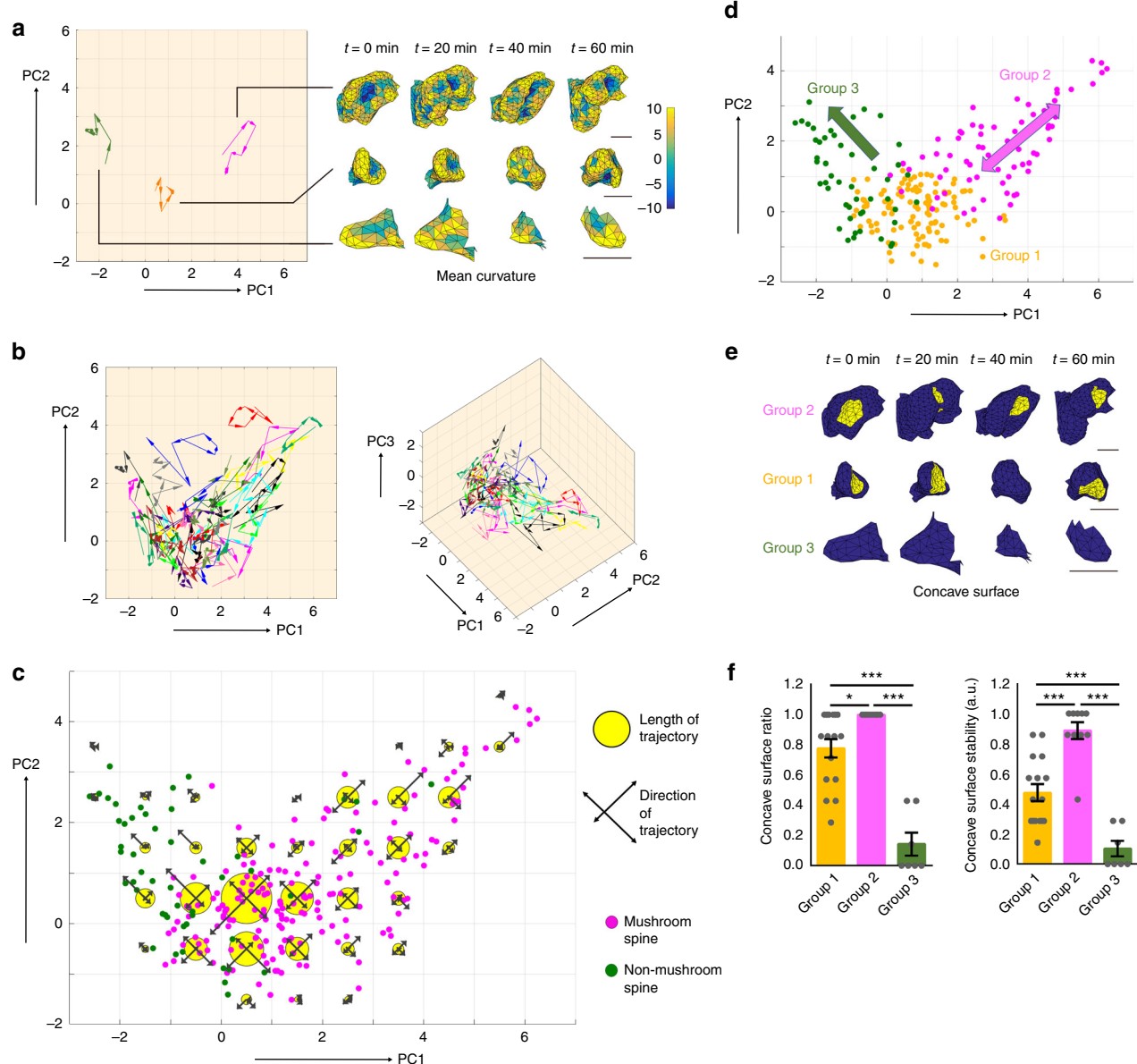

**Fig. 3** Spine shape transition studied by SIM-based geometrical analysis. **a** Trajectories of shape transition for three examples of spines mapped into the feature space. The right images of reconstructed spine shapes are from three spines at multiple time points (0–60 min). The values of mean curvatures are mapped onto the surface. Bar: 500 nm. **b** Projection and 3D maps of trajectories created from a population of spines (46 spines from eight neurons in five independent culture preparations). **c** The direction and length of trajectories in spine shape changes, mapped into the feature space. Total lengths of trajectories are shown as the radii of yellow circles, and trajectories projected onto four orthogonal directions are shown as black arrows. The PC1–2 plane is shown. The positions of spines at different time points were also mapped after SVM-based shape classification into mushroom and non-mushroom spines (magenta and green dots). **d** A scheme of three types of spine behavior in the feature space. Orange, magenta, and green dots indicate spines classified in groups 1, 2, and 3, respectively. **e** Mapping of the most concave surface on spine heads classified in groups 1, 2, and 3. Group 2 spines maintained the concave surface, whereas the concave surface was less stable in group 1, and did not exist in group 3. **f** The fraction of time points when the concave surfaces can be mapped to spine heads (concave surface ratio), and the fraction of time points when the concave surfaces located in the same direction within the spine heads (concave surface stability) were measured for three groups of spines [$n = 16$, 10, and 7 spines for groups 1, 2, and 3, respectively; one-way ANOVA followed by Tukey–Kramer procedures for multiple comparison tests; concave surface ratio: $F(2,30) = 41.57$, $p = 2.25 \times 10^{-9}$; concave surface stability: $F(2,30) = 33.78$, $p = 2.08 \times 10^{-8}$; *$p < 0.05$, ***$p < 0.001$. Data are presented as the mean ± SEM]

Time-lapse 3D-SIM images before and after uncaging revealed stabilization of the concave surface on the spine head (Fig. 4b). To evaluate the size of the concave surface, we calculated the volume difference between the convex hull and the spine head, normalized by the spine head volume (Fig. 4c). This index (concave volume ratio) reflects the relative size of the concave surface on the spine head, and its increase lasted more than an hour after the induction of synaptic plasticity (Fig. 4d).

The increase in the size of a spine's concave surface may indicate an increase in the junctional area between the presynaptic and postsynaptic components. Synaptic cell adhesion molecules are involved in the formation of synaptic junctional

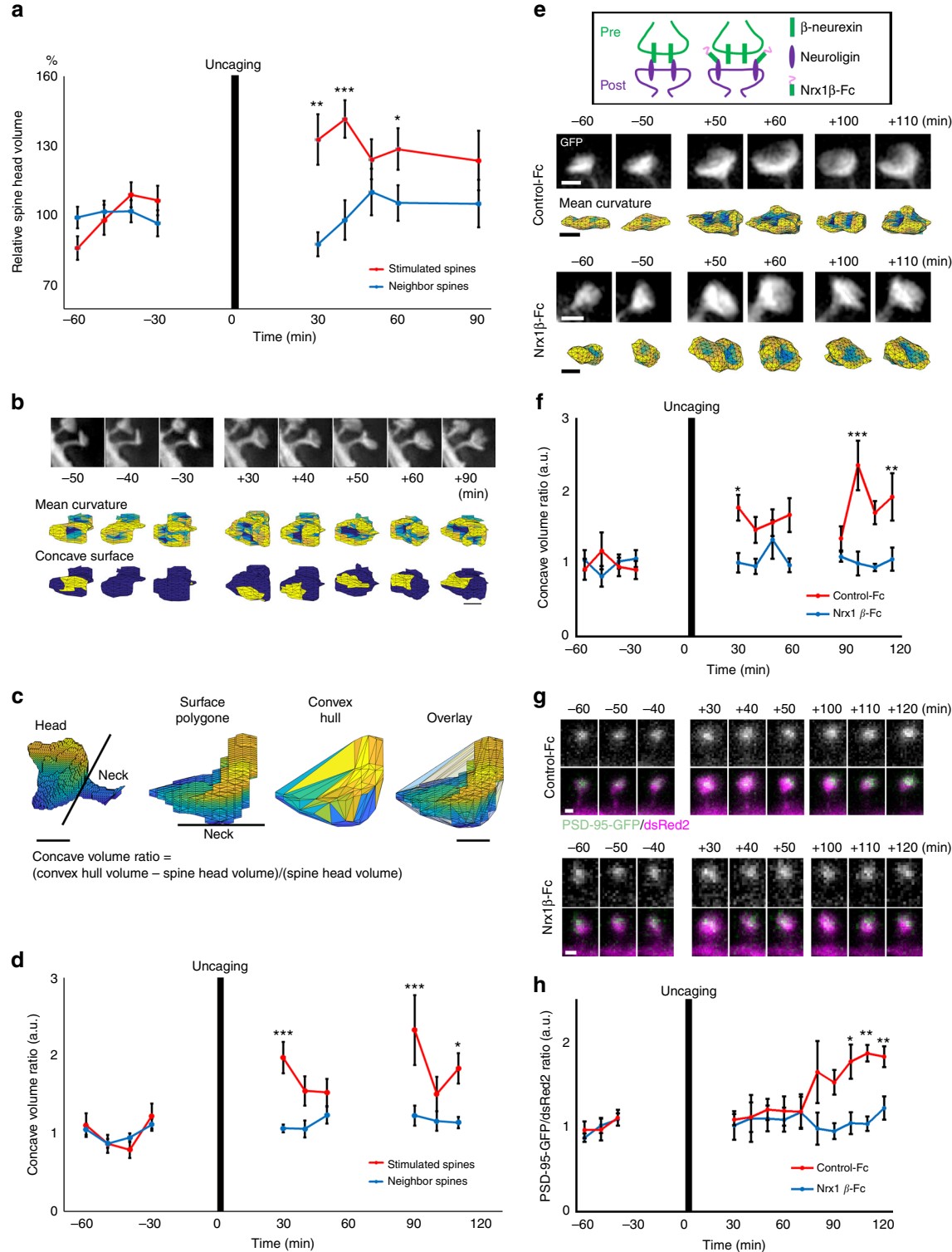

structures. Trans-synaptic interaction between the neurexin and neuroligin is important for formation of excitatory synapses[24,25] and is essential for synaptic plasticity[26]. Moreover, the extra-cellular domains of these proteins can form a superstructure resembling the extracellular structure of the synaptic cleft[27]. Among the four neuroligin isoforms in rodents, neuroligin 1 localizes at excitatory postsynaptic sites[28] and plays a dominant role in hippocampal long-term potentiation[26]. Because the affinity of neurexin 1β for neuroligin 1 is significantly higher

than that of other neuroligin isoforms[29], we expected that exogenous application of neurexin 1β-IgG fusion protein (neurexin 1β-Fc) would effectively and transiently block neuro-ligin 1 function (Fig. 4e). This manipulation inhibited enlarge-ment and stabilization of the concave surface, supporting the idea that membrane adhesion mediated by neuroligin 1 is involved in increasing the junctional surface of spines (Fig. 4f). This neuroligin-dependent mechanism of stabilization of the post-synaptic concave surface may be independent from the activity-

**Fig. 4** Plasticity-related changes in spine head geometry. **a** Time course of changes in spine head volume after glutamate uncaging [one-way ANOVA; stimulated: $n = 14$ spines; neighbor: $n = 14$ spines; *$p < 0.05$, **$p < 0.01$, ***$p < 0.001$]. **b** Structural changes in spine head before and after glutamate uncaging. SIM projection images of spines are shown in the upper row. Surface mapping of mean curvature and detected concave surfaces are shown in the middle and lower rows. Bar: 500 nm. **c** Images of reconstructed spine surface polygon with or without spine neck, convex hull mesh, and their overlay. Volume difference between the spine head and the convex hull was divided by the spine head volume, and this value is presented as the concave volume ratio for the graphs in panels (**d**) and (**f**). Bar: 500 nm. **d** Measurement of the volume difference between entire spine heads and the convex hull before and after glutamate uncaging. The graph shows the prolonged increase in this value after glutamate uncaging. [One-way ANOVA; stimulated: $n = 6$ spines; neighbor: $n = 10$ spines; *$p < 0.05$, ***$p < 0.001$]. **e** Inhibition of plasticity-associated changes in spine head geometry by neurexin 1β-Fc. SIM projection images of spines are shown in the upper row for both control-Fc and neurexin 1β-Fc conditions. The images in the lower rows show surface mapping of mean curvature. Bar: 500 nm. **f** Measurement of the volume difference between entire spine heads and the convex hull before and after glutamate uncaging in the presence of neurexin 1β-Fc. [One-way ANOVA; control-Fc: $n = 6$ spines; Nrx1β-Fc: $n = 6$ spines; *$p < 0.05$, **$p < 0.01$, ***$p < 0.001$]. **g** Inhibition of PSD-95 accumulation in the late phase of spine structural plasticity by neurexin 1β-Fc. Images of PSD-95-GFP are shown in the upper rows, and images of PSD-95-GFP merged with dsRed2 are shown in the lower rows. Bar: 500 nm. **h** Inhibition of PSD-95-GFP fluorescence increase after glutamate uncaging by neurexin 1β-Fc. [One-way ANOVA; control-Fc: $n = 4$ spines; Nrx1β-Fc: $n = 5$ spines; *$p < 0.05$, **$p < 0.01$]. Details of statistics in this figure is provided in Supplementary Table 2

dependent neuroligin cleavage, which rapidly regulates presynaptic function via the neurexin[30].

The time course of spine structural plasticity has been proposed to consist of three phases[31]. After the initial phase of rapid actin reorganization, on the order of several minutes, the actin cytoskeleton is stabilized for about an hour, followed by slow accumulation of PSD proteins. Because the postsynaptic concave surface is maintained for more than an hour after uncaging, and neuroligin 1 directly binds the prominent PSD scaffolding protein PSD-95[25], we next investigated whether blockade of neuroligin 1 function also affects the delayed accumulation of PSD-95. Consistent with previous reports[31], the amount of PSD-95 in spines was unaltered within the initial 60 min, but began to increase ~80 min after uncaging (Fig. 4g, h). We found that application of neurexin 1β-Fc effectively blocked the delayed increase of PSD-95 (Fig. 4g, h). Together, these results suggest that the postsynaptic concave surface stabilized by neuroligin 1 is an important structural element for the consolidation of synaptic plasticity.

In summary, we developed a method of reliably reconstructing and measuring the surface geometry of dendritic spines from 3D-SIM images. The data regarding spine volume, area, and length can be converted to absolute values after calibration with mesh data obtained from EM images. The method is efficient enough to perform data acquisition and quantitation of more than a 1000 spines within several days, which cannot be achieved by reconstruction of EM data. The method could be modified for the analysis of images obtained by other super-resolution techniques, such as stimulated emission depletion (STED) microscopy, which is suitable for the analysis of dendritic spines in the tissue environment[7,32,33]. Although the lateral resolution of 3D-SIM is lower than that of STED microscopy, the superior axial resolution of 3D-SIM is advantageous for reconstruction of spine geometry. Ideally, isotropic 3D-STED imaging combined with the technique of the surface mesh reconstruction should be applied to achieve higher-resolution measurement of spines[11].

We identified the concave surface of spine heads, which interact with the presynaptic terminal, as a unique structure that is expanded and stabilized by plasticity-inducing signals. We propose that this surface structure plays important roles in actin-dependent shape changes by recruiting regulatory molecules of actin filament nucleation and branching[34]. A recently identified interaction between neuroligin and the WAVE complex may play a role in increased actin-dependent shape changes after expansion of the concave spine surface[35]. This model suggests that concave spine surface that persists after glutamate uncaging serves as a core structural element of synaptic plasticity.

## Methods

**Neuronal culture from genetically modified mice**. ICR mice (Japan SLC), heterozygous synGAP mutant mice (synGAP$^{+/−}$)[15], homozygous knock-in mice harboring a knock-in of a kinase-dead K42R mutation in Ca$^{2+}$/calmodulin-dependent protein kinase IIα (CaMKIIα$^{K42R/K42R}$)[16], and Thy1-H$^{YFP/+}$ mice were used in this study. All animal experiments were approved by the animal welfare ethics committee of the University of Tokyo.

Dissociated hippocampal cultures were prepared from E16.5 ICR mouse embryos[36]. Briefly, hippocampi were treated with trypsin (Gibco) and DNase (SIGMA), and then mechanically dissociated. Cell suspension in the MEM containing B18 supplement, L-glutamine (Gibco), and 5% FCS (Equitech-Bio) was plated onto a glass-bottom dish (MatTek, #1.5) coated with poly-L-lysine (SIGMA). Two days after plating, 5 μM ara-C (SIGMA) was added to prevent glial cell proliferation.

Hippocampal dissociated cultures from synGAP$^{+/−}$ mice or CaMKIIα$^{K42R/K42R}$ mice were prepared using the same basic culture protocol as in the experiments using wild-type ICR mice, except that hippocampi from each embryo were dissociated separately. For genotyping of dissociated cultures, a piece of tissue from each embryo was saved prior to dissection of hippocampi. Genomic DNA purification was performed with the QuickGene DNA tissue kit (WAKO), and genotypes were determined by PCR using either High-Speed DNA polymerase (Kaneka) or ExTaq (TAKARA) following standard protocols provided by the manufacturers. Primer sequences for genotyping were as follows:
For synGAP genotyping PCR:
Sense primer for WT: 5′-GTCAGTGGGACATGGAAGTAG-3′
Sense primer for mutant: 5′-CTTCCTCGTGCTTTACGGTATC-3′
Antisense primer (common): 5′-CTGATCAGCCTGTCAGCAATG-3′
For CaMKIIα K42R genotyping PCR:
Sense primer: 5′-GGTCTTGAAGACTGTCTGGTGTGAGA-3′
Antisense primer: 5′-CACAGGCCAGTTTAGGTCTTGCTAGG-3′
Cell suspensions were prepared separately from each embryo after genotyping and plated at densities between 1.2 and $1.8 \times 10^5$ cells per dish. The whole procedure for dissociated cell culture was completed within 2.5 h.

**Fluorescent protein expression and DiI labeling of neurons**. Ca$^{2+}$-phosphate transfection was performed after 8–9 days in vitro according to a standard procedure[37]. All fluorescent proteins were expressed under the control of β-actin promoter. Either a GFP expression vector, a mixture of GFP and PSD-95-TagRFP expression vectors, or a mixture of dsRed2 and PSD-95-GFP expression vectors in calcium chloride solution was combined with the same volume of 2 × phosphate buffer and incubated at 25 °C for 15 min to generate a calcium phosphate–DNA co-precipitate. Culture medium was replaced with transfection medium containing the precipitate, and cells were incubated for 50 min in a 5% CO$_2$ incubator at 37 °C. The composition of transfection medium was identical to that of the culture medium, except that L-glutamine and FCS were omitted. Cells were returned to the original culture medium and incubated until the imaging experiments.

Differentiated single neurons in culture were labeled with 1,1′-dioctadecyl-3,3,3′,3′-tetramethylindocarbocyanine perchlorate (DiI; Molecular Probes)[38]. Fixed neurons in a culture dish were placed on the stage of an inverted microscope (IX71, OLYMPUS). DiI was dissolved in fish liver oil at saturating concentration and applied to individual cell bodies by pressure ejection with a FemtoJet (Eppendorf). The cells were left to stand for 30 min at 25 °C to allow the dye to spread, and then washed three times with PBS.

**Uncaging of caged glutamate**. Single-photon laser photolysis of 4-methoxy-7-nitroindolinyl-caged glutamate (MNI-Glu) was performed using a 405 -nm continuous-wave (CW) laser[39]. An uncaging laser (OBIS 405LX-100, Coherent)

was combined with an imaging line of a Nikon Structured Illumination Microscope system (N-SIM) using a dichroic mirror. The uncaging laser was controlled separately using the Coherent Connection software (Coherent) via transistor–transistor logic (TTL) generated by an Arduino UNO microcontroller. The uncaging laser was aligned to the center of the imaging field before each experiment.

For glutamate uncaging, hippocampal neurons were maintained at 37 °C in $Mg^{2+}$-free Tyrode's solution (119 mM NaCl, 2.5 mM KCl, 4 mM $CaCl_2$, 0 mM $MgCl_2$, 25 mM HEPES, and 30 mM glucose; pH 7.4) with 1 μM tetrodotoxin (TTX; WAKO) and 500 μM MNI-Glu (Tocris). Medium-size spines with clear heads and necks were selected for induction of structural change. Single-photon glutamate uncaging was performed by 2 msec pulses repeated at 1 Hz for 1 min with the center of the focused laser beam 1–2 μm away from the tip of the spine[39]. Precise control of the sample position was achieved by operating a motorized XY stage with N-SIM encoders. Laser intensity was set to 0.05–0.15 mW at the back aperture of the objective lens.

**Sample preparation for SIM imaging.** After 18–22 days in vitro, hippocampal neurons were washed with PBS and fixed with 4% paraformaldehyde and 0.5% glutaraldehyde in PBS for 30 min at 25 °C. Samples after transfection of GFP expression plasmids were mounted in Prolong® Diamond (Molecular Probes). DiI-labeled neurons were imaged in PBS. For correlative light microscopic and EM observation, cells were washed with PBS and fixed with 2% paraformaldehyde and 2% glutaraldehyde in PBS for 30 min at 25 °C.

**Apparatus for SIM imaging.** SIM imaging was performed with a N-SIM based on an inverted microscope (ECLIPSE Ti-E, NIKON), equipped with an oil immersion TIRF objective lens (SR Apo TIRF 100×, N.A. 1.49, NIKON), a laser system consisting of 405, 488, 561, and 640-nm diode lasers (LU-NV, NIKON), and an EMCCD camera (iXon3 DU-897E, Andor Technology). SIM imaging with this system is based on a previous report[17]. Briefly, excitation lasers were coupled to a multimode optical fiber, collimated, and directed to a fused silica linear transmission-phase grating. A shutter in an intermediate pupil plane discarded all diffraction orders except for 0 and ± 1. The three beams were refocused in the back focal plane of the objective lens. The beams produced as diffraction orders + 1 and −1 were focused near the opposing edges of the back focal plane aperture, and the beam produced as order 0 was focused at its center. Three-dimensional data were acquired with five-pattern phases spaced by $2\pi/5$ and three-pattern orientations spaced 60° apart. The acquired images were computationally reconstructed to obtain a high-resolution image with resolutions of ~115 nm in the x- and y-dimensions and ~270 nm in the z-dimension.

**SIM image acquisition.** Prior to SIM imaging, the temperature of the microscope system and specimens was stabilized at 28–29 °C to minimize position and aberration fluctuation. Spherical aberration induced by refractive index mismatch was corrected for each sample by adjustment of an objective correction collar. The EMCCD camera, which has a 512 × 512 pixel array consisting of 16-μm square pixels, was operated in read-out mode at 1 MHz with 16 bit analog-to-digital conversion with EM gain. An image stack with its size of 7.56 μm in the z dimension was acquired at 63 axial planes with 120 nm z-steps that satisfied the Nyquist criterion requirements. Dendritic segments that were isolated from unintended signals, such as axons or dendrites from other neurons, were carefully selected, because objects brighter than targeted dendritic segments were problematic in threshold setting for SIM image reconstruction.

**SIM image reconstruction.** All image processing steps were performed in three dimensions. The acquired datasets, comprising 63 axial sections of 512 × 512 pixels, were computationally reconstructed using reconstruction stack algorithm V2.10 of NIS-Elements AR (NIKON). The voxel size of the reconstructed images was 32 nm in the x- and y-dimensions and 120 nm in the z-dimension, with 16 bit depth. 3D-SIM data were validated using the SIMcheck plugin for ImageJ[40,41] (Supplementary Fig. 1). First, the standard SIMcheck procedure was performed with raw SIM data and the reconstructed image. Second, using a standalone Fourier transform plugin of SIMcheck, 2D fast Fourier transform (FFT) was applied to each slice of the reconstructed SIM image. The plugin, which was set to operate without a cutoff function and a window function (6% width), generated an 8 bit log-scaled (amplitude$^2$) Fourier power spectrum that was the same as the default FFT function of ImageJ for the 32 bit reconstructed data.

**Dual-color SIM imaging.** Hippocampal neurons expressing GFP were fixed with 4% paraformaldehyde and 0.5% glutaraldehyde in PBS for 30 min at 25 °C, and non-transfected neurons were stained with DiI. Contact sites between GFP-expressing axons and DiI-labeled dendrites were identified, and large mushroom spines with putative contact sites with axons were imaged. Neurons expressing GFP and PSD-95-TagRFP were fixed with 4% paraformaldehyde and 0.5% glutaraldehyde in PBS for 30 min at 25 °C and imaged in PBS.

Prior to dual-color 3D-SIM imaging, image registration between channels was performed according to the standard protocol of NIS-Elements AR. The two

channels of the 3D-SIM images were sequentially acquired at the same z-position, and this step was repeated with multiple z-positions. The final imaging volume spanned a thickness of 5.4 μm, comprising 45 axial planes separated by 120-nm z-steps. The two channels of the image stacks were reconstructed in parallel using reconstruction stack algorithm V2.10 of NIS-Elements AR.

**Live SIM imaging.** Live cell imaging was performed after 18–22 days in vitro. Cells in the culture medium were placed in a heater stage system (INUG2H-TIZSH, Tokai Hit) at 37 °C with a continuous flow of 5% $CO_2$ to maintain the pH of the medium. A custom-made lid for the glass-bottom dish was utilized to minimize evaporation of the culture medium. During live imaging, the z-position was maintained by a perfect focus system (NIKON).

For live 3D-SIM, an image stack 7.56-μm thick was acquired, consisting of 63 axial planes separated by 120-nm z-steps. The series of 3D-SIM images were acquired every 10 min for a total period of up to 1 h, and the total exposure number was 6615 (15 patterns × 63 axial planes × 7 time points) at most. Excitation laser power was set to be minimal but sufficient to reconstruct images with a sufficient signal-to-noise ratio even at the end of the time-lapse series (Supplementary Fig. 14). Bleaching of GFP fluorescence may be attenuated by fast diffusion of GFP within the dendritic cytoplasm. Data quality of 3D-SIM imaging sets was confirmed using the SIMcheck plugin for ImageJ[40]. Acquisition of a single SIM image stack took 490 s (100 ms exposure time × 15 patterns × 63 axial planes + z-stage settle time). The acquisition speed of N-SIM was mainly limited by rotating and laterally translating the grating. The 3D-SIM volume at each time point was independently reconstructed using reconstruction stack algorithm V2.10 of NIS-Elements AR. Neurons were confirmed as remaining alive for at least 1 day after time-lapse imaging.

Prior to glutamate uncaging, live 3D-SIM images were obtained every 10 min in the culture medium in a volume 3.48-μm thick (29 axial planes separated by 120-nm z-steps). Subsequently, the culture medium was replaced with $Mg^{2+}$-free Tyrode's solution containing 1 μM TTX and 500 μM MNI-Glu, and an uncaging laser was applied to spines that protruded horizontally from dendritic shafts. After uncaging, the external solution was replaced with the original culture medium for subsequent live 3D-SIM imaging. For neuroligin blocking experiments, neurons were first imaged in the culture medium. Glutamate uncaging was performed in $Mg^{2+}$-free Tyrode's solution containing 1 μM TTX, 500 μM MNI-Glu, and either 50 μg/ml recombinant human nexrin 1β-Fc (without splice insert 4, R&D Systems) or 50 μg/ml recombinant human Fc (R&D Systems) as a control. After uncaging, the external solution was replaced with the original culture medium containing 50 μg/ml recombinant human nexrin 1β-Fc or control-Fc, followed by time-lapse 3D-SIM imaging.

For imaging of PSD-95-GFP, N-SIM was operated in the wide-field mode. Hippocampal neurons expressing PSD-95-GFP and dsRed2 were examined after 18–22 days in vitro. Spines that protruded horizontally from dendritic shafts were imaged, and single-photon glutamate uncaging was performed. To avoid possible photo-bleaching of PSD-95-GFP, the output power and focus position of the uncaging laser were carefully adjusted. For neuroligin blocking experiments, the culture medium was replaced with $Mg^{2+}$-free Tyrode's solution containing 1 μM TTX, 500 μM MNI-Glu, and either 50 μg/ml recombinant human nexrin 1β-Fc or 50 μg/ml recombinant human Fc as a control. Live imaging was performed subsequently at multiple time points before and after glutamate uncaging.

**Correlative light microscopic and EM observation.** Correlative light and electron microscopy was performed with dissociated neuronal culture[42]. Following SIM imaging, phase contrast images of the same neurons were recorded and used as a reference to obtain EM images of the identical dendritic segments. For transmission electron microscopy, samples were post-fixed with 1.0% $OsO_4$ and 1.5% potassium ferrocyanide in 0.1 M cacodylate buffer, and stained with 1.0% tannic acid in 0.05 M cacodylate buffer. Samples were dehydrated and embedded in epoxy resin (Poly/Bed®812 Luft Formulations, Polysciences). After 2 days of curing at 60 °C, the area imaged by SIM was cut out, and the bottom glass was removed by treating the block with hydrogen fluoride. After trimming the block, ultra-thin sections of 50–60 nm thickness were prepared in a Reichert ultramicrotome with a diamond knife and mounted in Formvar-coated copper slot grids. TEM images were acquired at 80 keV on a transmission electron microscope (JEM-1010, JEOL) with a CCD camera (TemCam-F216, TVIPS) at ×5000 magnification. Image alignment and reconstruction were performed using the Reconstruct software package (SynapseWeb).

Two operators manually identified the area (about 0.25 μm$^2$) with the largest negative curvature from 3D-SIM data of each spine head. The positions of the synaptic junctional areas were recorded independently from the 3D-reconstructed EM images. The positions of the smallest negative curvature and the synaptic junctional area were compared and judged to determine whether the two areas overlapped. The positions were matched in 9 mushroom-shaped spines out of 10.

**Tissue preparation and laser scanning confocal microscopy.** Male transgenic mice (Thy1-H$^{YFP/+}$, 3 months old) were deeply anaesthetized and perfused with 4% paraformaldehyde in 0.1 M phosphate buffer. Brains were removed, post-fixed in 4% paraformaldehyde for 6–8 h, and then sectioned with 40 μm thickness in the

coronal plane on a vibratome (VT-1000S, Leica). The sections were washed with 88% (weight by volume) histodenz (SIGMA) in 0.1 M phosphate buffer and were mounted directly on coverslips (high-tolerance coverglass D = 0.17 ± 0.005 mm, Matsunami) in 88% histodenz for imaging.

Confocal microscopy was performed with an A1 confocal laser scanning microscopy system (NIKON). An oil immersion TIRF objective lens (Apo TIRF 100×, N.A. 1.49, NIKON) was used, and images were collected with confocal aperture of 0.5 AU. The image stacks with their size of 30.7 µm in $x$–$y$ plane and 7.56 µm in the $z$ axis were acquired. The voxel size of the images was 30 nm, 30 nm, and 120 nm in the $x$, $y$, and $z$ directions, respectively. Prior to imaging, the temperature of the microscope system and specimens was confirmed to be stabilized, and spherical aberration was corrected for each sample by adjustment of an objective correction collar.

**Automated image thresholding and surface mesh generation**. Two independent thresholding methods were used to automatically isolate dendrites and spines from SIM image stacks. The reconstructed SIM image stacks were first processed by multilevel thresholding based on Otsu's method, and the resultant binary images were further processed by geodesic active contours to refine object boundaries.

Otsu's method automatically searches the threshold that maximizes the between-class variance of pixel intensity[43]. Direct application of Otsu's method for thresholding SIM images of dendrites was not successful, mainly because the images contained both large objects with strong fluorescence (dendrites) and small objects with weak fluorescence (spines). However, Otsu's method was previously extended for multilevel thresholding, and we found that the modified method could reliably detect multiple thresholds for both the strong fluorescence signal of dendrites and the weak signal of spines (Supplementary Fig. 2). After generation of binary images by multilevel thresholding, the resultant binary images were further processed to refine the boundaries of dendrites and spines by geodesic active contours[18]. The technique is based on active contours (snakes) evolving in time and pulled toward object boundaries until the energy function reaches its minimum. The evolving contours can split and merge in the iterative process, and this property helps to eliminate and merge isolated image pixels below the resolution of SIM microscopy (Supplementary Fig. 2). MATLAB has built-in functions for both multilevel thresholding [multithresh()] and geodesic active contours [activecontour()]. A custom MATLAB script was developed for processing SIM image stacks with these two image processing techniques (Supplementary Software "SIM_activecontour").

The binary image stack generated by thresholding SIM images was processed for automated detection of spines (Supplementary Fig. 3). A custom MATLAB script was developed for spine detection and polygon mesh generation (Supplementary Software "SIM_spine_detection"). Individual spines were detected and isolated automatically (Fig. 1a, Supplementary Figs. 3 and 4). Dendritic shafts were fitted with elliptic cylinders, and voxel clusters outside of the best fit of elliptic cylinders were identified as spine candidates (Supplementary Fig. 4b). These spine candidates were further sorted by criteria of their volumes and shape characteristics. We first rejected spine candidates with their volumes close to the resolution limit of SIM imaging. For this purpose, objects with their volumes less than twice the volume of a rectangular cuboid with its edge lengths equal to the theoretical resolutions of SIM images (< 0.01 µm³) were rejected (Supplementary Fig. 4b). We next rejected objects that were judged to be too elongated along the axis of the dendritic shaft. For objects larger than 0.32 µm³, we first calculate the largest cross-section (OCS) in the plane normal to the longitudinal axis of the parent dendritic shaft and the length of the object along the longitudinal axis of the dendritic shaft (OL). If the ratio of OCS to OL was <4, the object was judged to be too elongated and rejected. For objects smaller than 0.32 µm³, we rejected objects with their OCS/OL ratio <2. This size-dependent adjustment of the threshold is based on the fact that small spines tend to have lower OCS/OL ratios (small stubby spines tend to have larger bases connected to dendritic shafts). The shapes of virtual objects classified by these criteria are shown in Supplementary Fig. 4c.

The junction between spines and dendritic shafts was identified as follows. First, we dilated the spine with two voxels (~60 nm) toward the junction and isolated the overlap between the dendritic shaft and the dilated spine as the shaft-side surface (blue pixels in the right image of Supplementary Fig. 4b). Next, the spine voxels within 150 nm from the voxel centroids of the shaft-side surface were identified (orange pixels in the right image of Supplementary Fig. 4b) as the spine-side surface. We finally eliminated the voxels of the spine-side surface. This procedure is effective in removing the voxels that stretch on the shaft surface away from the spine base. On the other hand, this procedure removes a small number of voxels in the spine base close to the shaft surface. We confirmed that this voxel elimination procedure does not largely affect the subsequent spine measurement procedure by comparing the outputs with the manual measurement (The ranges of errors were < 60 nm for spine length, <0.5 µm² for spine area, <0.1 µm³ for spine volume. All errors were <12% of the measured values.).

The isolated spine voxels were further processed by the marching cube algorithm for isosurface triangulation. In the subsequent analyses, the mesh data were saved as polygon file format (.ply). Alternatively, the polygon meshes of dendritic segments were loaded into an open-source 3D computer graphics software (Blender, the Blender Foundation) for detection and isolation of polygon

meshes for dendritic spines. Filopodia-like protrusions were not excluded from the analyses. In some cases (< 15% of the total spine population), polygon meshes of a pair of nearby spines were merged; these were omitted from the analyses.

**Spine geometrical analysis**. Three-dimensional triangular mesh surfaces of spines were processed using custom MATLAB scripts (Supplementary Software "Geometric_calculation" and "Geometric_curvature"). First, basic geometrical parameters, including spine length, spine surface area, and spine volume, were calculated using the following equations[20]. Schematic explanations of the measured values are provided in Supplementary Fig. 5.

Spine length ($L$) was calculated using the following equations:

$$L = \frac{1}{n}\sum_{i=1}^{n}\left(\overrightarrow{|\mathbf{A}_i|}\right) \tag{1}$$

$$\overrightarrow{\mathbf{A}_i} = \left(x_i - c_x, y_i - c_y, z_i - c_z\right) \tag{2}$$

where $i$ stands for the index of all vertices with distances from the centroid of the spine/shaft junctional plane ($c_x, c_y, c_z$) larger than the upper 95% of the distances for all vertices. ($x_i, y_i, z_i$) are the coordinates of the vertices. (Note that this parameter is different from the authentic curvilinear length from the base to the tip of the spine. Our parameter gives more reasonable estimates for spines with complex morphology, which have multiple protrusions and extending thin edges, but provides lower estimates for long curved spines.)

Spine surface area ($S$) was calculated using the following equation:

$$S = \sum_i \frac{1}{2}\sqrt{(v_{i1}w_{i2} - w_{i1}v_{i2})^2 + (w_{i1}u_{i2} - u_{i1}w_{i2})^2 + (u_{i1}v_{i2} - v_{i1}u_{i2})^2} \tag{3}$$

where $i$ stands for the index of all elementary triangles. ($u_{i1}, v_{i1}, w_{i1}$) and ($u_{i2}, v_{i2}, w_{i2}$) are vectors corresponding to two edges of triangle $i$.

Spine volume ($V$) was calculated using the following equation:

$$V = \sum_i \frac{1}{6}(-x_{i3}y_{i2}z_{i1} + x_{i2}y_{i3}z_{i1} + x_{i3}y_{i1}z_{i2} - x_{i1}y_{i3}z_{i2} - x_{i2}y_{i1}z_{i3} + x_{i1}y_{i2}z_{i3}) \tag{4}$$

where $i$ stands for the index of all elementary triangles. ($x_{i1}, y_{i1}, z_{i1}$), ($x_{i2}, y_{i2}, z_{i2}$), and ($x_{i3}, y_{i3}, z_{i3}$) are the coordinates of the vertices of triangle $i$.

Additional geometric parameters that reflect more complex morphological features were also included. Schematic explanations of the measured values are provided in Supplementary Fig. 5.

Convex hull volume (CHV) is volume of the smallest convex set of vertices that contains the spine polygon meshes. Vertices and volume of the convex hull for a given spine mesh can be calculated by the built-in MATLAB function convhulln (). From CHV, convex hull ratio (CHR) was calculated as follows:

$$\text{CHR} = (\text{CHV} - V)/V \tag{5}$$

Average distance (AD), the average distance between the individual vertices and the centroid of the spine/shaft junctional plane, was calculated using the following equation:

$$\text{AD} = \frac{1}{N}\sum_{i=1}^{N}\left(\overrightarrow{|\mathbf{A}_i|}\right) \tag{6}$$

where $i$ and $N$ stand, respectively, for the index of all vertices and their total number. Coefficient of variation in distance (CVD) is coefficient of variation of the calculated distances between the individual vertices and the centroid of the spine/shaft junctional plane.

Open angle (OA) is the average angle formed by the spine axis and each vertex vector. (A vertex vector starts from the centroid of the spine/shaft junctional plane and ends at a vertex. Spine axis is specified by the average of all vertex vectors.) Mushroom spines with flat spine heads, as well as stubby spines, have larger values of OA. OA was calculated using the following equations:

$$\text{OA} = \frac{1}{N}\sum_{i=1}^{N}\left\{\cos^{-1}\left(\frac{\overrightarrow{\mathbf{M}} \cdot \overrightarrow{\mathbf{A}}_i}{|\overrightarrow{\mathbf{M}}| \cdot |\overrightarrow{\mathbf{A}}_i|}\right)\right\} \tag{7}$$

$$\overrightarrow{\mathbf{M}} = \frac{1}{N}\sum_{i=1}^{N}\overrightarrow{\mathbf{A}}_i \tag{8}$$

where $i$ and $N$ stand, respectively, for the index of all vertices and their total number.

The third group of parameters associated with spine morphology involves surface geometry. Curvatures on discrete surfaces made by polygon meshes can be estimated by the following operators[21].

Mean curvature (MC) at the vertex ($x_i$) specified by the vector ($\overrightarrow{x_i}$) was calculated using the discrete approximation of the Laplace-Beltrami operator K:

$$K(x_i) = \frac{1}{2A} \sum_{j \in n} \left( \cot \alpha_{ij} + \cot \beta_{ij} \right) \left( \overrightarrow{x_i} - \overrightarrow{x_j} \right) \quad (9)$$

$$MC = \frac{1}{2} |K(x_i)| \quad (10)$$

where $n$ is a group of vertices surrounding the central vertex $x_i$, $A$ is the size of the barycentric region with its center at $x_i$, $x_j$ is the $j_{th}$ vertex surrounding the center vertex $x_i$, and $\alpha_{ij}$ and $\beta_{ij}$ are the opposite angles of the two triangles sharing the edge $ij$.

Gaussian curvature (GC) at the vertex ($x_i$) was calculated using the following equation:

$$GC = \frac{1}{A} \left( 2\pi - \sum_{j \in n} \theta_j \right) \quad (11)$$

where $n$ is a group of vertices surrounding the central vertex $x_i$, $\theta_j$ is the $j_{th}$ angle of the incident triangles at the central vertex $x_i$, and $A$ is the size of barycentric region with its center at $x_i$.

MC and GC were calculated for all vertices belonging to spine surface meshes. Averages of MC and GC (avMC and avGC) were used as morphological parameters for individual spines. For the quantitative analysis of local surface curvature, spine mesh sizes were increased to match the resolution of SIM (the minimum area of triangle meshes was >0.004 μm$^2$).

In summary, we calculated 10 parameters ($L$, $S$, $V$, CHV, CHR, AD, CVD, OA, avMC, and avGC) for each spine as descriptors of spine shape features.

**PCA and SVM**. Geometrical parameters of spines were calculated and analyzed by the PCA method. From our initial analysis of independence among 10 descriptors ($L$, $S$, $V$, CHV, CHR, AD, CVD, OA, avMC, and avGC), we selected five descriptors ($L$, $V$, CHR, CVD, and OA) based on two criteria. First, we selected two descriptors that reflect principal structural features (length and volume). Second, three other descriptors that were relatively independent with the initial two descriptors and with each other (the averages of pairwise correlation coefficients were <0.3) were selected. PCA was applied to two types of image samples, neurons expressing GFP and neurons labeled by DiI[44]. In both cases, the first three components in the reduced representation covered more than 94% of the variance. Comparison of the weights for GFP and DiI projection matrices indicated that the differences were small (5.6 ± 4.1%, mean ± SD, for weights >0.4). Based on this observation, whether the projection matrix generated from the merged data of GFP and DiI could be used for efficient dimensionality reduction was next investigated. When the two datasets were merged, the GFP dataset was converted to DiI data using the coefficients determined from the data obtained from neurons imaged by both GFP and DiI (Supplementary Fig. 9). Again, the first three PCs after PCA covered >90% of the variance in the data. This projection matrix was used in subsequent spine analyses for neurons from genetically modified mice and time-lapse 3D-SIM experiments. For this purpose, the class sklearn.decomposition.PCA in scikit-learn, a machine-learning library (scikit-learn.org), was used.

For classification of mushroom and non-mushroom spines with SVM[45], one expert manually labeled 1335 and 914 polygon meshes of spines generated from 3D-SIM images of GFP-expressing and DiI-labeled neurons, respectively. Using these labeled datasets, two hyperparameters, $C$ (penalty parameter) and $\gamma$ [coefficient for radial basis function (rbf) kernel], were optimized by grid search, and the model performance was assessed by k-fold cross-validation (Fig. 2c). The trained SVM classified mushroom spines of GFP-expressing and DiI-labeled neurons with identical accuracy (89%). When the trained SVMs for GFP and DiI were exchanged, the high accuracy was still preserved (85% for classifying GFP data with a SVM trained with DiI data; 87% for classifying DiI data with a SVM trained with GFP data). Because the distinction between mushroom and non-mushroom spines is not always unambiguous, and classification by human operators is prone to high variability, three additional experts were introduced for manual labeling of 400 polygon meshes of spines from GFP-expressing neurons. Comparison of the classifications by the three experts and SVM revealed that the percentage match between human operators and SVM was comparable with that among human operators, implying that SVM-based classification of spines is advantageous because it removes human subjectivity (Supplementary Table 1). Hyperparameter tuning and SVM classification were performed using classes sklearn.model_selection.GridSearchCV and sklearn.decomposition.SVC in scikit-learn.

**Analysis of spine shape transition**. Automated image thresholding and surface mesh generation were applied independently for each time point. After image processing, the difference in threshold values for Otsu's method from the average were confirmed to be within 20%. Corresponding spines at different time points were further analyzed using the same parameters for spine geometrical analysis. Trajectories of spine shape transition over time were mapped in the feature space following PCA (Fig. 3b). Vectors of shape transitions in the feature space were mapped into grids with edge lengths of $1 \times$ SD, and the total lengths of trajectories

and the vector components projected onto orthogonal directions were calculated (Fig. 3c). Classification of spines after time-lapse imaging was based on SVM-based classification into mushroom and non-mushroom spines, followed by 2D mapping to the plane, with the abscissa representing position in PCA feature space and the ordinate representing the orientation preference of the trajectories (Supplementary Fig. 15b). The orientation preference was calculated by the ratio between two orthogonal components of the shape trajectories in the plane of PC1 and PC2. Two orthogonal directions were chosen to be either parallel or perpendicular to the line 45° counterclockwise from the PC1 axis. When the orientation preference was >1, the trajectory was judged to be closer to the direction from small/thin to large/round spines (Fig. 3c, d).

Mapping of the concave surface of spine heads was based on MCs calculated at spine vertices. As a first step, up to 10 vertices with the smallest negative MC values were selected. Next, the average values of MCs were calculated over two-ring neighborhoods (a given start vertex plus the first and second neighborhood vertices), the area of which roughly corresponds to the size of the PSD, which has a diameter of 400 nm. Based on the average MCs for the 10 candidate areas, the area with the smallest negative average MC was selected and mapped to the spine head. When the smallest negative MC of a single vertex within the area was greater than −7 or the number of vertices with negative MC values was less than 4, this surface was rejected, and the spine was judged to have no clearly identifiable concave surface. This method reliably predicted the positions of pre- and postsynaptic interfaces in the correlative SIM and EM analysis when prominent concave surfaces were detected (Fig. 1c, Supplementary Figs. 6 and 7). Concave surface ratio was calculated as the number of time points, when the spine was judged to have a clear concave surface divided by the number of total imaging time points for that spine. Stability of concave spine surfaces in time-lapse experiments (Fig. 3f) was evaluated by relative movement of the center vertex ($V_c$) in the concave surface at two time points ($t_1$ and $t_2$). The lines that passed the centroid of whole-spine polygon ($C_{spine}$) and the $V_c$ were drawn at $t_1$ and $t_2$, and the angle between these two lines after alignment of $C_{spine}$ was calculated [∠$V_c(t_1)C_{spine}V_c(t_2)$]. If this angle was >60°, the concave surface was judged to be unstable. Concave surface stability was calculated as the number of time points, when the spine was judged to maintain a stable concave surface divided by the number of total imaging time points minus 1.

Structural changes in concave surfaces before and after induction of spine structural plasticity were evaluated by measuring CHV and $V$ of the spine heads and calculating CHR. Larger CHR indicates more space between the convex hull and original spine volume, which mainly reflects the volume made by the concave surface of the spine.

**Model spine generation and simulation**. For simulation of spine image degradation by optically induced blur, in silico model spines were generated using a custom MATLAB script. Initial parameters were set as spine head radius of 500 nm, spine neck radius of 150 nm, and spine neck length of 700 nm. We also removed a part of spine head volume to mimic concave surface on the spine head. The volume removed from the spine head was defined by the overlap of two spheres, one for spine head volume and the other for exclusion volume, with its radius of 400 nm. The distance between the centers of two spheres was set to 450 nm. Model spines with different sizes were generated by proportionally transform the original spine shape. The model spine volume was set in a range of 0.08–0.57 μm$^3$.

The image stacks were convolved with the Gaussian filters corresponding to SIM resolution (115 nm, 115 nm, and 270 nm in the $x$, $y$, and z directions, respectively) or resolution of confocal microscopy with confocal aperture set to 0.5 AU (190 nm, 190 nm, and 410 nm in the $x$, $y$, and z directions, respectively) by the built-in MATLAB function convn (). The convolved image stacks were first binarized by multilevel thresholding based on Otsu's method without geodesic active contours, followed by conversion to polygon mesh data. MC was calculated at individual spine vertices. If clustered vertices of negative curvature (<−2.5) were detected on a spine head, the spine image degraded by the optical system was judged to still preserve their original concave surface.

**Comparison of spine shape in culture and in vivo**. We compared our own spine volume data in cultured neurons expressing GFP ($n = 1335$) with the data of EM-reconstructed in vivo spines ($n = 938$) available in the open data depository linked to the publication by Bloss et al.[23]. Both data were generated from hippocampal pyramidal neurons. The data for spine length in vivo were also taken from a previous publication ($n = 100$)[46]. Previous reports of direct comparison between chemical fixation and cryofixation reported that tissue shrinkage in the process of fixation, dehydration, and embedding for EM sample preparation was ~26% in the neuropil[47]. We assumed that the shrinkage of spines should be proportional to the overall shrinkage of the neuropil. Based on this idea, we reduced the measured spine volumes in cultured neurons by the factor of 0.74 before generating the histogram shown in Supplementary Fig. 12. It should be noted that in the case of chemical fixation of cultured neurons, shrinkage of dendrites and spines was negligible (Supplementary Fig. 14). This difference in the effect of shrinkage may be derived from the difference in the concentration of fixatives, the speed of chemical reaction, or the osmolarity of fixative solutions[47,48].

**Statistics**. Data are expressed as means ± SEM unless otherwise noted. The statistical tests used for each experiment and the exact value of $n$ (number of spines) are indicated in the corresponding figure legends. Statistical significance was determined by one-way ANOVA followed by Tukey–Kramer procedures for multiple comparison tests, using the Statistics and Machine-Learning Toolbox™ of MATLAB (MathWorks). The equality of probability distributions was evaluated by Kolmogorov–Smirnov test. $P$-values < 0.05 were considered statistically significant (*$p < 0.05$, **$p < 0.01$, ***$p < 0.001$).

**Reporting summary**. Further information on experimental design is available in the Nature Research Reporting Summary linked to this article.

## Code availability

Custom scripts written in MATLAB R2017b are provided as Supplementary Software, and custom scripts written in Python are available from the corresponding author upon request.

## Data availability

All data files are available from the corresponding author on reasonable request.

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

## Acknowledgements

We thank Y. Aizawa, K. Muranaga, and K. Ohkubo for supporting the preparation of dissociated neuronal culture. This study was supported by Grants-in-Aid for Scientific

Research (17H01387 and 18H04727 to S.O.), Core Research for Evolutional Science and Technology from the Japan Science and Technology Agency (JPMJCR14W2 to S.O.), the Project for Elucidating and Controlling Mechanisms of Aging and Longevity from the Japan Agency for Medical Research and Development (17gm5010003 to S.O.), and the UTokyo Center for Integrative Science of Human Behavior (CiSHuB). S.G.N.G. and N.H.K. are supported by Medical Research Council and the Japan Science and Technology Agency Strategic Cooperation Program.

## Author contributions

Y.K. and S.O. designed the research; Y.K, T.H., K.O. and Y.S. performed the experiments; N.H.K. and S.G.N.G. contributed to analysis of mutant animals; Y.K. and S.O. analyzed data; and Y.K. and S.O. wrote the paper.

## Additional information

**Competing interests:** The authors declare no competing interests.

**Journal Peer Review Information:** *Nature Communications* thanks Valentin Nagerl, Simon Rumpel, and other anonymous reviewer(s) for their contribution to the peer review of this work. Peer reviewer reports are available.

