## [Peer Review File · Nature Communications]

Reviewers' comments:

Reviewer #1 (Remarks to the Author):

The manuscript uses structured illumination microscopy in combination with computational analysis to measure and analyze the morphology of dendritic spines in dissociated neurons from primary cultures. They demonstrate that their innovative approach can pick up subtle changes in spine morphology induced by experimental manipulations (genetic mutations, glutamate uncaging), and make the point that it facilitates the analysis of a large number of dendritic spines in a relatively rapid and objective way.

Even though many software programs for spine analysis have been developed over the years, they are still far from perfect, usually not making good use of the morphological information contained in microscopy images, especially those based on super-resolution techniques.

The paper attempt to address this important and unmet need in the field by applying recent computational approaches to improve the detection and classification of morphological features.

However, I remain reserved about the novelty and advance that the current version of the study presents. The SIM images certainly are very beautiful and the correlative SIM/EM is amazing, but that is not the point of the study. 'Machine learning' is an exciting key word, but it is not clear, what the concrete benefits are over existing approaches. Machine learning usually relies on 'supervised learning' with a test data set, where the true answer is known. However, in the case of spines there is no ground truth regarding their classification, i.e. there is no independent marker to distinguish the different spine types (stubby, thin, mushroom, filopodial) and their continuum of intermediate shapes.

Main criticism:

1- The analysis is used on dissociated neurons, where optical resolution and image contrast are much better than in more intact preparations. But dissociated neurons are not really an interesting preparation anymore when it comes to investigating subtle nanoscale alterations in spine morphology, because of the artificial conditions. The authors should attempt to acquire SIM images in brain slices (which would add novelty in terms of the microscopy application) and would allow them to test their analysis program on data sets with more realistic image quality. Alternatively, they could also challenge their program by adding noise to the data and see how robustly it performs. Or they could also use slice data from the literature based on super-resolution (STED) microscopy.

2- The authors should analyze spine necks, which should be properly resolved by SIM, including how they are affected by the experimental manipulations.

3- In the introduction, the authors should provide a concise summary of the existing spine analysis programs, detailing their strong and weak points, to help the reader understand how their new software improves on the current method.

4- In general, the authors should discuss more extensively the limitation of their approach. For example, some very short spines are sometime difficult to separate from small bump on the denritic shaft. How the approach deal with this configuration ?

5- The manuscript is sometime hard to read (examples: mentioning the animal model and the brain region observed at the beginning of the results would be helpful; the PC1-3 parameters used in figure 1b, 2a...are not clearly defined in the results section; etc). The author should try to clarify a bit the

result part.

6- Their data seem sometime under analyzed. Indeed, while the power of their approach is to quantify spine morphology, quantification is sometime missing and the manuscript is a bit fuzzy. For example:

- Results - 1st section: "in a majority of large spines". What Percentage? What are large spines?
- Same section: "five descriptors that were relatively independent". What is there criteria?
- Results – 3rd section: "most group 2s pine". How much?
- etc

7- Finally, the author never discuss how the measured morphological parameters compare with the literature. Indeed, even if some parameters such as the curvature has never been properly discussed, lots of other (spine volume, length, neck diameter...) has been extensively discussed in the literature in all type of preparation (dissociated cultured, slice, in vivo) using different technique (EM, optical microscopy...).

Reviewer #2 (Remarks to the Author):

This paper by Kashiwagi et al details a method for the analysis of dendritic spine morphology. This uses structured illumination microscopy and computation of the images to make accurate 3d models from which geometric analysis can be carried out.

The automated analysis is able to detect dendritic spines, and automatically record their volume, surface area, length and degree of curvature. The paper tests this method by observing differences in spine populations from different genotypes and as a result of glutamate uncaging.

The paper is well written, and clearly laid out, and demonstrates the effectiveness of such an approach to spine shape analysis. The main criticism of this piece of work, however, is that the description does not address the fact that the system being imaged is a dissociated culture. In such a preparation there is the maximum opportunity to obtain high-quality images, but in one that is a long way from brain tissue. The classification of spine types, though interesting, looks very different from the ultrastructural analyses of brain tissues, and particularly those in the hippocampus. I am concerned that the range of spine shapes shown in this work is only relevant to such a culturing method. While this should not detract from the methodology used it does raise the question as to the relevance of such a fine-grained analysis. Another example is shown in the EM image of Figure 1C. This reveals a remarkable amount of extracellular space around a synapse. An amount that would never be seen in vivo.

The authors include some correlative light and electron microscopy to verify the geometrical analysis. They use serial section transmission electron microscopy to reconstruct a segment of dendrite that was imaged with SIM. Two minor questions arise from this part of the work. The authors describe how their method of analysis is able to accurately reconstruct spines that may lie above or below the dendritic shaft as their resolution in the z is high enough that the structures are not merged. However, this does not seem to be tested with the electron microscopy as all spines appear to be emerging laterally from the dendrite. Also, it's not clear if the electron microscopy is carried out on dendritic spines that were imaged live with SIM or after fixation. As some of the SIM imaging is carried out on aldehyde-fixed material and some on live cells the question arises as to how well the correlation is maintained? In other words, does the fixation alter the geometry in any way? As the paper emphasizes the way in which the concave surfaces can be analyzed it would be useful to know if this parameter is altered in any way during the fixation and EM processing.

The details of the methodology are given and illustrated. However, it's not entirely clear from the description of the parameters, volume, surface area, and length, as to exactly what part of the spine this corresponds. Figure 1b shows an image but does not indicate which part of this spine is measured to give the parameters that are indicated in the Figure. At what point do the spine stop and the dendrite start? For the length measurement, between which two points is this recorded. The mathematical equations are given but it would be helpful if this could be illustrated on some example models.

Reviewer #3 (Remarks to the Author):

The manuscript by Kashiwagi et al. describes a novel and highly interesting approach to study the dynamics of dendritic spines, the morphological correlates of excitatory synapses. The study has high merit, both on the technological as well as the biological side, by applying structured illumination microscopy, advanced image and data analysis to various genetic models of synaptic mutants. In summary, I think this study will be of high interest to a large number of neuroscientists.

Nevertheless, there are a few points that caught my attention, I think that should be addressed prior a possible publication, and which are mentioned in the order as they appear in the manuscript:

Page 3, para 1: citation 4 is mentioned to make the case of a continuous distribution of morphological parameters. This point is actually made in later from the same group (Loewenstein et al., 2015, J. Neurosci.).

Page 5, para 1: Here, the authors describe their data processing pipeline, in which a PCA is applied to the higher-dimensional data before classification using a SVM. I think here the authors could better justify this step. PCA and other dimensionality reduction methods are useful for illustration of high-dimensional data, typically at the cost of throwing away some information. The SVM could be also applied to the original data, even if some of the variables are not independent. To me it did not become clear why this step was included.

Page 5, para 2: The authors talk about 'resolution gain' on the three dimensions using their method. Maybe I missed this: where do they actually give the concrete resolution of the microscope? I think this would be a good part of the manuscript to provide the reader with this important information.

Page 7, para 2: 'generalized spine head dominance': Here maybe the authors could better explain what this is meant with this, I found this term not very intuitive.

Page 8, para 1: Here the authors write that 'mushroom spines were clustered in the feature space', which is contradicting their first sentence of the para, where they write that the feature space was continuous and did not feature signatures of potential subclasses or categories of spines. And indeed Fig. 2a shows a continuous probability distribution of spine features that is monotonically falling off from the center in all directions. I could not spot any evidence of 'clumping' or 'clustering' of a set of particular feature combinations. I think for the purpose of analyzing their data, it is fine to operationally define subcategories and to train a classifier to set a boundary in a distribution, where no 'natural' boundary can be found. However, my concern is that the way the manuscript is written here, it can be easily confused with the notion, that there is indeed a subcluster, a true category for mushroom spines. Of course, as the feature space is designed in a meaningful manner, similar looking spines (as different examples of 'mushroom spines') will be projected close together, but I would

simply avoid the terms 'cluster' or 'category' in this context, as this implies indeed a structure in the distribution of spine morphologies that apparently is not in the data.

Responses to reviewers

Reviewer #1

We would like to thank the Reviewer #1 for valuable comments. We followed the suggestions and revised the manuscript based on additional experiments and data analysis.

The manuscript uses structured illumination microscopy in combination with computational analysis to measure and analyze the morphology of dendritic spines in dissociated neurons from primary cultures. They demonstrate that their innovative approach can pick up subtle changes in spine morphology induced by experimental manipulations (genetic mutations, glutamate uncaging), and make the point that it facilitates the analysis of a large number of dendritic spines in a relatively rapid and objective way.

Even though many software programs for spine analysis have been developed over the years, they are still far from perfect, usually not making good use of the morphological information contained in microscopy images, especially those based on super-resolution techniques.

The paper attempt to address this important and unmet need in the field by applying recent computational approaches to improve the detection and classification of morphological features.

However, I remain reserved about the novelty and advance that the current version of the study presents. The SIM images certainly are very beautiful and the correlative SIM/EM is amazing, but that is not the point of the study. 'Machine learning' is an exciting key word, but it is not clear, what the concrete benefits are over existing approaches. Machine learning usually relies on 'supervised learning' with a test data set, where the true answer is known. However, in the case of spines there is no ground truth regarding their classification, i.e. there is no independent marker to distinguish the different spine types (stubby, thin, mushroom, filopodial) and their continuum of intermediate shapes.

We appreciate the reviewer for pointing out the importance of showing the benefit of using machine learning-based spine classification. We understand the reviewer's opinion that spine classification into stubby, thin, mushroom, and filopodial is dependent on subjective criteria and there is no ground truth. Indeed, that is the reason that we restrict application of machine learning in classification of spine subtypes in the analysis of genetically modified

mice. In this context, the most important question is if our technique can effectively and objectively classify spines into subgroups and this subgrouping help understand the subtle structural changes in spine shape induced by genetic mutations. As we could successfully detect spine shape changes in specific subgroups of spines in both synGAP heterozygote mice and CaMKIIalpha kinase-dead mice, SIM-based quantitative analysis combined with machine learning-based classification was proved to be useful and efficient in objective identification of spine structural alterations in mutant mice. We believe that our approach has wide application in objective analysis of spine pathology in animal models for neurological and psychiatric disorders.

Additionally, we want to stress that even if ground truth does not exist and classification is arbitrary, still “standardization” of spine classification is very useful. At present, most scientific publications showing changes in spine morphology rely on manual spine classification. Manual spine classification depends on resolution of imaging system, ways of determining spine surface, and judgement by researchers. The technique we presented in this manuscript utilized imaging system with high correlation to electron microscopic reconstruction, objective and standardized method of spine shape determination, and objective machine learning-based classification that is reasonably consistent with judgements of multiple researchers. By facilitating other synapse laboratories to share this standardized approach and deposit the data in an open website, we expect that this approach can be a real breakthrough in spine research through data standardization and sharing.

Main criticism:

1- The analysis is used on dissociated neurons, where optical resolution and image contrast are much better than in more intact preparations. But dissociated neurons are not really an interesting preparation anymore when it comes to investigating subtle nanoscale alterations in spine morphology, because of the artificial conditions. The authors should attempt to acquire SIM images in brain slices (which would add novelty in terms of the microscopy application) and would allow them to test their analysis program on data sets with more realistic image quality. Alternatively, they could also challenge their program by adding noise to the data and see how robustly it performs. Or they could also use slice data from the literature based on super-resolution (STED) microscopy.

As suggested by the reviewer, we performed additional experiments for application of spine geometry analysis to intact tissue preparations (see the section “Computational geometry of

spines in vivo” in the main text). First, it is important to investigate the relationship between theoretical and practical resolution of different imaging systems with different types of samples and applicability of our computational approach. For this purpose, we performed in silico analysis of spine morphology. We created model spines, which had concave surfaces on spine heads, and convoluted the model with Gaussian filters. Gaussian functions match the theoretical resolutions of SIM and confocal laser scanning microscopy with a confocal aperture of 0.5 AU, which improves resolution in comparison with conventional confocal microscopy. The model images after convolution were processed by our computational geometrical analysis and then judged if they preserved their original nanoscale details or not. Using this technique, we could estimate the range of spine head sizes in which either SIM or confocal microscopy can detect the concave surface. The results indicate that confocal microscopy with confocal aperture of 0.5 AU can detect concave surfaces on spines with their volume larger than $0.18 \mu\text{m}^3$ (described in the main text lines 215-228, Supplementary Fig. 11a-c).

The spines of which volumes were more than $0.18 \mu\text{m}^3$ correspond to 25% of the total spines and 61% of them were judged to have concave surfaces. The remaining 75% spines (smaller than $0.18 \mu\text{m}^3$) showed concave spine surface much less frequently (13%). Thus, imaging methods that can detect concave surface of spines with their volumes larger than $0.18 \mu\text{m}^3$ are practically useful. Our model-based analysis indicates that, not only SIM, but also confocal microscopy with a small confocal aperture should be effective in detecting spine nanoscale features.

Based on this in silico experiment, we next performed imaging of hippocampal slices both SIM and confocal microscopy with a 0.5 AU confocal aperture and analyzed the images with our computational geometrical technique. In real data sets, we confirmed that spine nanoscale features, such as concaved surface on spine heads, can be detected in tissue slices, with both SIM and confocal microscopy with a small confocal aperture (described in the main text lines 229-236, Supplementary Fig. 11d-f).

Finally, we collected spine population data by confocal microscopy of intact hippocampal slices and applied the PCA and SVM-based classification of mushroom spines. The distribution of spine data after PCA was highly similar between samples from cultured neurons imaged by SIM and intact tissue imaged by confocal microscopy (described in the main text lines 237-253, Supplementary Fig. 13). The performance of SVM-based classification of mushroom spine (88.0% of accuracy) was comparable to that of data obtained from cultured neurons (89.3% of accuracy). In addition, the size distribution of in vivo spines measured by confocal microscopy [$0.081 \pm 0.091 \mu\text{m}^3$ (Mean \pm SD, $n = 165$)] was comparable to that of in vitro spines measured by SIM [$0.079 \pm 0.078 \mu\text{m}^3$ (Mean $245 \pm$

SD, n = 1,335)] and also that from previous publication of in vivo spines with EM reconstruction [$0.076 \pm 0.082 \mu\text{m}^3$ (Mean \pm SD, n = 938)] (Bloss et al., Nat. Neurosci. 2018). From these analyses, we concluded that our computational geometrical analysis can be applied to brain slices with realistic image quality of scanning microscopy, which is more suitable for tissue imaging than SIM.

2- The authors should analyze spine necks, which should be properly resolved by SIM, including how they are affected by the experimental manipulations.

We agree with this reviewer that spine neck morphology is an important structural feature of spines. On the other hand, spine neck analysis is not straightforward from several reasons.

►First, there is no consensus in the definition of spine neck. The border between spine head and neck is arbitrary and may change by the criteria. Even if we try to set a specific definition of the border between spine head and neck, reliable definition is difficult to find. A substantial fraction of mushroom spines show gradual change in the diameter of spines and the clear border does not exist.

►Second, measurement of spine neck width is difficult for thin spine neck. Due to the resolution limit of SIM image, the measured size of spine neck width becomes less accurate for thin necks, especially in the vertical direction.

►Third, we want to restrict our structural parameters to those that can be applied to all of the spines. Spine necks can be defined only in mushroom-type spines. These category-specific structural parameters are difficult to handle in computational analysis.

From these reasons, we did not directly use spine neck-related structural features as descriptors in our analysis. It should be emphasized that our strategy is to use structural parameters that can be measured in all spines reliably within the limit of SIM resolution. After measurement of these multiple reliable parameters, we could indirectly extract more difficult structural features, such as spine neck morphology. We believe this strategy is more effective and sound, compared with measuring a subset of spines for arbitrary-defined spine neck morphology at the border of the resolution limit.

Beside the above discussion about our strategy of spine analysis, it should be valuable and informative for the readers if we can provide experimental data that illustrate to what extent spine neck structure can be accurately imaged by the SIM technique. For this purpose, we performed comparative analysis of spine neck width between SIM images and EM

reconstruction from the same samples. The results indicate that reliable measurement is possible above the width of 150 nm in XY plane and 350 nm in Z direction. The result is reasonable under the resolution limit of SIM imaging. This point was described in the main text (lines 148-154) and in Supplementary Fig. 8d.

3- In the introduction, the authors should provide a concise summary of the existing spine analysis programs, detailing their strong and weak points, to help the reader understand how their new software improves on the current method.

We thank this reviewer for suggesting a concise summary of the existing spine analysis program in the introduction. We added the summary of previous spine analysis program (lines 55-69).

4- In general, the authors should discuss more extensively the limitation of their approach. For example, some very short spines are sometime difficult to separate from small bump on the dendritic shaft. How the approach deal with this configuration ?

We thank this reviewer for this suggestion. We recognize the limitation of our approach and added the discussion about problems associated with detection of very short spines. Our spine detection program automatically detects spine-like structures with a rule implemented in the algorithm. Therefore, detection of spine candidate is objective and reproducible. In this program, we reject spine-like objects with their volume less than 0.01 cubic micrometer. This threshold is derived from the idea that this volume roughly corresponds to twice the volume of cubic with its edge length equal to the resolution limit and objects with their volume less than this threshold may fall short of their identity as clear protrusions. We also rejected objects that are judged to be too elongated along the axis of the dendritic shaft. The details of the judgement process were added to the main text (lines 109-117) and Methods section (lines 252-282).

5- The manuscript is sometime hard to read (examples: mentioning the animal model and the brain region observed at the beginning of the results would be helpful; the PC1-3 parameters used in figure 1b, 2a...are not clearly defined in the results section; etc). The author should try to clarify a bit the result part.

We appreciate this reviewer for pointing out these problems. We added explanations for the animal models and the brain region observed at the beginning of the results (lines 86-87,

lines 94-97).

We also explained the definition of PC1-3 in the results. Basically, principal components are normalized linear combinations of the original descriptors and in a strict sense, they can be defined only by mathematical equations. In our case, however, PC1-3 could be interpreted as structural properties of spines. This point was discussed in the main text of the revised manuscript with reference to the distribution of spines with typical morphologies in Fig. 2a and b (lines 164-171).

6- Their data seem sometime under analyzed. Indeed, while the power of their approach is to quantify spine morphology, quantification is sometime missing and the manuscript is a bit fuzzy.

We appreciate this reviewer for pointing out these problems. We modified the text according to the suggestions as follows.

- Results - 1st section: “in a majority of large spines”. What Percentage? What are large spines?

As suggested, we changed the sentence with the actual numbers. We found that the concave surface was present in 61% of spines of which volumes were more than $0.175 \mu\text{m}^3$. This spine population corresponds to 25% of the total spines and 84% of them are classified as mushroom spines using our machine learning method. This point was added to the main text (lines 135-138).

- Same section: “five descriptors that were relatively independent”. What is there criteria?

As suggested, we added the description on the evaluation of independence among descriptors. We first selected two descriptors that reflect principal structural features (length and volume) and then selected three other descriptors that were relatively independent from the first two descriptors and also with each other (the averages of pair-wise correlation coefficients were less than 0.3). This point was added to the main text (lines 157-162).

- Results – 3rd section: “most group 2s pine”. How much?

As suggested, we changed the sentence to include actual numbers. This is based on the

finding that 10 out of 11 spine time lapse images showed the maintenance of concave surface for 1 h. The main text was modified to include this point (lines 287-288).

7- Finally, the author never discuss how the measured morphological parameters compare with the literature. Indeed, even if some parameters such as the curvature has never been properly discussed, lots of other (spine volume, length, neck diameter...) has been extensively discussed in the literature in all type of preparation (dissociated cultured, slice, in vivo) using different technique (EM, optical microscopy...).

We analyzed volume distribution of spines quantitated by the SIM-based method in comparison with EM-reconstructed spines in vivo, both are on hippocampal pyramidal neurons. For this comparison, two factors should be taken into account. First, lower resolution in SIM image extends spines in Z direction and induces overestimation of spine structure. This effect can be estimated from the correlative imaging experiments in Fig. 1e, where structural parameters (length, area, and volume) of spines from EM and SIM images were compared. From this experiment, we estimated the true spine volume was 75% of the volume measured by SIM. Second, tissue shrinkage in the process of fixation, dehydration, and embedding for EM sample preparation should be considered. Direct comparison of chemical fixation and cryo-fixation reported volume shrinkage of the neuropil was 26%. We incorporated these two factors in our comparison and converted the spine volume of cultured neurons based on SIM to that of in vivo spines in EM samples. Using these conversion parameters, we next compared our SIM-based measurement with previous reports of spine size using EM-based reconstruction. We found that spine volumes of two different preparations were remarkably similar. Spine volume of cultured hippocampal neurons measured by SIM-based method was $0.079 \pm 0.078 \mu\text{m}^3$ (Mean \pm SD, n = 1,335) and spine volume of CA1 pyramidal neurons measured by EM reconstruction was $0.076 \pm 0.082 \mu\text{m}^3$ (Mean \pm SD, n = 938). By similar analysis of spine length, we found that spine length of cultured neurons measured with SIM was $1.01 \pm 0.44 \mu\text{m}$ (Mean \pm SD, n = 1,335) and spine length of hippocampal CA1 neurons measured by EM reconstruction was $0.95 \pm 0.42 \mu\text{m}$ (Mean \pm SD, n = 100). The results indicate similar distribution of spine volumes in culture and in vivo. This analysis is also consistent with our measurement of in vivo spine volume based on confocal microscopy (Supplementary Fig. 11 and 13). This comparison is now presented as Supplementary Fig. 12 and mentioned in the main text (lines 224-226, lines 243-246).

Comparison of spine neck diameter was not carried out in this study due to the limit of optical

resolution of SIM technique. The measured size of spine neck width by SIM is less accurate for thin spine necks, as mentioned previously. We discussed this point in the main text (lines 148-154) and presented the data in Supplementary Fig. 8d.

Reviewer #2

We would like to thank the Reviewer #2 for valuable comments. We followed the suggestions and revised the manuscript based on additional experiments and data analysis.

This paper by Kashiwagi et al details a method for the analysis of dendritic spine morphology. This uses structured illumination microscopy and computation of the images to make accurate 3d models from which geometric analysis can be carried out.

The automated analysis is able to detect dendritic spines, and automatically record their volume, surface area, length and degree of curvature. The paper tests this method by observing differences in spine populations from different genotypes and as a result of glutamate uncaging.

The paper is well written, and clearly laid out, and demonstrates the effectiveness of such an approach to spine shape analysis. The main criticism of this piece of work, however, is that the description does not address the fact that the system being imaged is a dissociated culture. In such a preparation there is the maximum opportunity to obtain high-quality images, but in one that is a long way from brain tissue. The classification of spine types, though interesting, looks very different from the ultrastructural analyses of brain tissues, and particularly those in the hippocampus. I am concerned that the range of spine shapes shown in this work is only relevant to such a culturing method. While this should not detract from the methodology used it does raise the question as to the relevance of such a fine-grained analysis. Another example is shown in the EM image of Figure 1C. This reveals a remarkable amount of extracellular space around a synapse. An amount that would never be seen in vivo.

This reviewer concerns the possibility that the structure of spines developed in dissociated culture system may differ from that of in vivo spines. We took this criticism seriously and first analyzed volume distribution of spines quantitated by the SIM-based method in comparison with EM-reconstructed spines in vivo, both are on hippocampal pyramidal neurons. The details of this comparison are described in the responses to the point 7 of First Reviewer. The results indicate that spines in dissociated culture system and those in vivo show very similar size distribution. The volume distribution was also consistent with the data obtained by confocal microscopy of in vivo dendritic spines (lines 237-253 of the main text,

also see our responses to the point 1 of First Reviewer). This point is mentioned in the main text (lines 224-226 of the main text) and the data was presented in Supplementary Fig. 12. The important finding in this manuscript is presence of concave surface on the head of large spines. By using the technique of high resolution confocal microscopy, we could confirm the presence of similar concave surface on spine heads of hippocampal neurons in fixed brain tissue (lines 229-236 in the main text, Supplementary Fig. 11d-f). Concave surface in spine heads could be also detected in recently published STED images of hippocampal neurons in tissue sections (eLIFE, 2018, 7:e34700). Therefore, nanoscale characteristics we found in cultured neurons should be informative in studies of spine nanostructure in intact tissue.

STED image of hippocampal neuron dendrite. Concave surface can be detected (arrows).
Figure 2 of eLIFE, 2018, 7:e34700

The authors include some correlative light and electron microscopy to verify the geometrical analysis. They use serial section transmission electron microscopy to reconstruct a segment of dendrite that was imaged with SIM. Two minor questions arise from this part of the work.

The authors describe how their method of analysis is able to accurately reconstruct spines that may lie above or below the dendritic shaft as their resolution in the z is high enough that the structures are not merged. However, this does not seem to be tested with the electron microscopy as all spines appear to be emerging laterally from the dendrite.

We had additional data for correlative analysis of SIM and EM reconstruction, which can be used for the evaluation of spines protruding above or below the dendritic shaft. Although the SIM images of these spines have less resolution in their longitudinal axes, they have better resolution in their cross-section. Therefore, the reconstructed images of spines protruding above or below the dendritic shaft are not necessarily worse than spine extending horizontally.

The representative three-dimensional images of SIM and EM are shown in Supplementary Figure 8, indicating similar morphological features of vertically extending spines. To evaluate the accuracy of length measurement for spines protruding from dendritic shaft with different angles, we grouped spines by the angles made by the spine long axes and the XY-plane of the SIM image, and then plotted the difference in spine length between SIM-based and EM-based measurements. This difference reflects the accuracy of SIM-based measurement. The data shows similar extent of the difference between groups, indicating reliable shape detection and measurement of vertically protruding spines (lines 145-147 in the main text, Supplementary Fig. 8c).

Also, it's not clear if the electron microscopy is carried out on dendritic spines that were imaged live with SIM or after fixation. As some of the SIM imaging is carried out on aldehyde-fixed material and some on live cells the question arises as to how well the correlation is maintained? In other words, does the fixation alter the geometry in any way? As the paper emphasizes the way in which the concave surfaces can be analyzed it would be useful to know if this parameter is altered in any way during the fixation and EM processing.

The electron microscopy is carried out on dendritic spines that were imaged with SIM after fixation. This point was described in the main text (lines 127-130). In an independent experiment, we directly compared the same spines before and after chemical fixation and confirmed chemical fixation did not change the spine geometry, including concave surface in the spine head (line 261-262 in the main text, Supplementary Fig. 14c).

The details of the methodology are given and illustrated. However, it's not entirely clear from the description of the parameters, volume, surface area, and length, as to exactly what part of the spine this corresponds. Figure 1b shows an image but does not indicate which part of this spine is measured to give the parameters that are indicated in the Figure. At what point do the spine stop and the dendrite start?

Our spine detection program automatically detects spine-like structures with a rule implemented in the algorithm. Therefore, detection of spine candidate is objective and reproducible. In this program, we first fit the dendritic shaft with an elliptic cylinder and the volume outside of the cylinder was judged to belong to spines. The details of isolating junctions between the spines and the parental dendritic shaft are described in the Methods (lines 252-269) and in the Supplementary Fig.4. It is necessary to fine tune parameters for

proper fitting of the shaft segment with an elliptic cylinder and this process was done automatically in the program. This point is explained in more detail in the Methods (lines 270-282).

For the length measurement, between which two points is this recorded. The mathematical equations are given but it would be helpful if this could be illustrated on some example models.

As suggested by the reviewer, we added example models of our spine measurement in the Supplementary Fig.5. Definitions of some parameters (spine length) are not identical to the conventional ways of measurements and these models will help readers to understand the meaning of parameters we measured.

Reviewer #3

We would like to thank the Reviewer #3 for valuable comments. We followed the suggestions and revised the manuscript based on new data analysis.

The manuscript by Kashiwagi et al. describes a novel and highly interesting approach to study the dynamics of dendritic spines, the morphological correlates of excitatory synapses. The study has high merit, both on the technological as well as the biological side, by applying structured illumination microscopy, advanced image and data analysis to various genetic models of synaptic mutants. In summary, I think this study will be of high interest to a large number of neuroscientists.

Nevertheless, there are a few points that caught my attention, I think that should be addressed prior a possible publication, and which are mentioned in the order as they appear in the manuscript:

Page 3, para 1: citation 4 is mentioned to make the case of a continuous distribution of morphological parameters. This point is actually made in later from the same group (Loewenstein et al., 2015, J. Neurosci.).

We thank this reviewer for pointing out this error. We corrected the text as suggested (lines 41-43).

Page 5, para 1: Here, the authors describe their data processing pipeline, in which a PCA is applied to the higher-dimensional data before classification using a SVM. I think here the authors could better justify this step. PCA and other dimensionality reduction methods are useful for illustration of high-dimensional data, typically at the cost of throwing away some information. The SVM could be also applied to the original data, even if some of the variables are not independent. To me it did not become clear why this step was included.

It has been reported that in some cases data preprocessing by PCA and other dimensional reduction techniques can improve the performance of classifiers. In our case, SVM without PCA can perform well, but if SVM without PCA trained by GFP expressing neurons is used for classification of Dil-stained neurons, the performance is slightly worse than SVM with PCA. We think that this is mainly due to the fact that features that are only useful for GFP-

expressing neurons are utilized by SVM without PCA. Another possibility is that the feature space created by PCA is more suitable for grouping in a range of shapes of non-linear kernel we tested for SVM. We included this discussion in the main text (lines 185-194).

Page 5, para 2: The authors talk about ‘resolution gain’ on the three dimensions using their method. Maybe I missed this: where do they actually give the concrete resolution of the microscope? I think this would be a good part of the manuscript to provide the reader with this important information.

In this sentence we just mentioned general property of SIM-based imaging and did not claim that our computational approach can further improve the resolution beyond the limit of SIM technology. To clarify this point, we modified our text (lines 99-102) and presented the data of diffraction-limit image of fluorescent beads in the Supplementary Fig. 1d.

Page 7, para 2: ‘generalized spine head dominance’: Here maybe the authors could better explain what this is meant with this, I found this term not very intuitive.

Along the axis of PC3, spines were aligned according to their relative sizes of spine head against total spine volumes. To make this point clearer, we modified the main text (lines 168-170).

Page 8, para 1: Here the authors write that ‘mushroom spines were clustered in the feature space’, which is contradicting their first sentence of the para, where they write that the feature space was continuous and did not feature signatures of potential subclasses or categories of spines. And indeed Fig. 2a shows a continuous probability distribution of spine features that is monotonically falling off from the center in all directions. I could not spot any evidence of ‘clumping’ or ‘clustering’ of a set of particular feature combinations. I think for the purpose of analyzing their data, it is fine to operationally define subcategories and to train a classifier to set a boundary in a distribution, where no ‘natural’ boundary can be found. However, my concern is that the way the manuscript is written here, it can be easily confused with the notion, that there is indeed a subcluster, a true category for mushroom spines. Of course, as the feature space is designed in a meaningful manner, similar looking spines (as different examples of ‘mushroom spines’) will be projected close together, but I would simply avoid the terms ‘cluster’ or ‘category’ in this context, as this implies indeed a structure in the distribution of spine morphologies that apparently is not in the data.

We thank this reviewer for pointing out the importance of accurate description about distribution of the spine population in the feature space. We modified our text and avoid describing spine distribution as clusters or categories (main text lines 176-179). We also changed other sentences in the text to correct this point.

REVIEWERS' COMMENTS:

Reviewer #1 (Remarks to the Author):

A duplicated reference should be removed (Danzl et al. Ref# 11 and 33)

They should consider citing Nagerl et al (PNAS 2008), which first reported cup-shaped hippocampal spine heads with STED in live slice cultures and Pfeiffer et al (eLife 2018) with STED in perfusion fixed brains.

Otherwise, the authors have done a very good job in answering my queries, and I recommend publication of the manuscript.

Reviewer #2 (Remarks to the Author):

This paper is improved from the previous version and the authors have made some efforts to incorporate suggested changes. However, two issues need to be addressed.

The first one relates to the precision of title and the abstract. This is a method that is able to measure the morphology of dendritic spines morphology of live dendrites in a dissociated culture and also tissue slices that have been cleared and optimally processed for light microscopy. They are able to show that the method works on time-lapse images of dendrites in their in vitro, dissociated culture system, but not on in vivo imaged dendrites. It would be important to specify this in the abstract at the very least.

The second issue concerns the shrinkage that may or may not occur during fixation. In the methods section beginning on line 465 the authors indicate that the fixation causes shrinkage of 26%. At the beginning of this paragraph, they state that they, ' analyzed the volume distribution of spines...'. What is the volume distribution in this case, and how was this measured? The authors' value of neuropil shrinkage I presume does not mean the shrinkage of the spines, but the tissue as a whole, so it's not clear to me why this is measured if they are only interested in the spines. In the supplementary information, the authors show that the spines do not change volume during the fixation process. Therefore, I am confused about the issue of chemical fixation and shrinkage from the data that is presented. Also, the authors compare spine volumes from EM studies and from their own measurements and come up with a value of 26%. However, previous studies have not shown a shrinkage of spines in this order, only in the neuropil. I apologize if this is confusingly written here, but it's very confusing in the paper, and I am sure this can be addressed more succinctly.

Reviewer #3 (Remarks to the Author):

Kashiwagi and colleagues have addressed all points adequately, that I had raised in my initial review. From my side I would favour publication of the manuscript in its current form.

One small comment, nevertheless: The label in Figure 1 a 'Spine autodetection' means that a spine has the ability to detect itself, which I believe is not what was intended. 'Automated spine detection' appears more adequate.

Responses to reviewers

Reviewer #1

A duplicated reference should be removed (Danzl et al. Ref# 11 and 33)

We thank this reviewer for pointing out this error. We removed the reference as suggested.

They should consider citing Nagerl et al (PNAS 2008), which first reported cup-shaped hippocampal spine heads with STED in live slice cultures and Pfeiffer et al (eLife 2018) with STED in perfusion fixed brains.

We thank this reviewer for this suggestion. We cited these articles in our paper.

Otherwise, the authors have done a very good job in answering my queries, and I recommend publication of the manuscript.

We thank this reviewer for this positive response.

Reviewer #2 (Remarks to the Author):

This paper is improved from the previous version and the authors have made some efforts to incorporate suggested changes. However, two issues need to be addressed.

The first one relates to the precision of title and the abstract. This is a method that is able to measure the morphology of dendritic spines morphology of live dendrites in a dissociated culture and also tissue slices that have been cleared and optimally processed for light microscopy. They are able to show that the method works on time-lapse images of dendrites in their in vitro, dissociated culture system, but not on in vivo imaged dendrites. It would be important to specify this in the abstract at the very least.

We thank this reviewer comment. We added the text according to the suggestions to the abstract as follows:

“Here, we describe an accurate method for measurement and analysis of spine morphology based on structured illumination microscopy (SIM) and computational geometry in cultured neurons.”

The second issue concerns the shrinkage that may or may not occur during fixation. In the methods section beginning on line 465 the authors indicate that the fixation causes shrinkage of 26%. At the beginning of this paragraph, they state that they, ‘ analyzed the volume distribution of spines...’. What is the volume distribution in this case, and how was this measured?

We thank this reviewer for pointing out the vagueness of our description on the way we compared two data sets. On one hand, we analyzed our own data obtained by the SIM-based method from cultured hippocampal neurons. On the other hand, we searched the previous publications of EM-based reconstruction of dendritic spines in the hippocampus in vivo. We noticed that the paper by Bloss et al. (Nat Neurosci 2018) is linked to the raw data of spine volumes and utilized this data set to generate the histogram shown in Supplementary Figure 12.

We also have to clarify that the estimated shrinkage rate of ~26% stated in the text is also derived from the previous literature (Korogod, et al. eLIFE 2015) and not based on our own

experiments. We applied this shrinkage rate to proportionally reduce the spine volume measured by the SIM-based method before making the histogram in Supplementary Figure 12.

To clarify the description in the section “Comparison of spine shape in culture and in vivo” in the Methods section (pages 48-49), we modified the text as follows (underlined sentences):

“We compared our own spine volume data in cultured neurons expressing GFP (n = 1335) with the data of EM-reconstructed in vivo spines (n=938) available in the open data depository linked to the publication by Bloss et al²³.” Both data were generated from hippocampal pyramidal neurons. The data for spine length in vivo were also taken from a previous publication (n = 100)⁴⁶. “Previous reports of direct comparison between chemical fixation and cryofixation reported that tissue shrinkage in the process of fixation, dehydration, and embedding for EM sample preparation was ~26% in the neuropil⁴⁷.” We assumed that the shrinkage of spines should be proportional to the overall shrinkage of the neuropil. Based on this idea, we reduced the measured spine volumes in cultured neurons by the factor of 0.74 before generating the histogram shown in Supplementary Fig. 12. It should be noted that in the case of chemical fixation of cultured neurons, shrinkage of dendrites and spines was negligible (Supplementary Fig. 14). This difference in the effect of shrinkage may be derived from the difference in the concentration of fixatives, the speed of chemical reaction, or the osmolarity of fixative solutions^{47, 48}.”

The authors' value of neuropil shrinkage I presume does not mean the shrinkage of the spines, but the tissue as a whole, so it's not clear to me why this is measured if they are only interested in the spines.

We again thank this reviewer to raise this important issue. Our basic assumption is that the shrinkage of spines should be proportional to the overall shrinkage of the neuropil. In the paper of Korogod et al. (eLIFE 2015), the volume of neurites was reported to decrease in the neuropil. As dendritic shafts and spines are major components of the neurites, proportional shrinkage of spines and the total neuropil should be a straightforward interpretation of their data. We described this point in the Methods section as follows (underlined sentences):

“We compared our own spine volume data in cultured neurons expressing GFP (n = 1335) with the data of EM-reconstructed in vivo spines (n=938) available in the open data

depository linked to the publication by Bloss et al²³. Both data were generated from hippocampal pyramidal neurons. The data for spine length in vivo were also taken from a previous publication (n = 100)⁴⁶. Previous reports of direct comparison between chemical fixation and cryofixation reported that tissue shrinkage in the process of fixation, dehydration, and embedding for EM sample preparation was ~26% in the neuropil⁴⁷. We assumed that the shrinkage of spines should be proportional to the overall shrinkage of the neuropil. Based on this idea, we reduced the measured spine volumes in cultured neurons by the factor of 0.74 before generating the histogram shown in Supplementary Fig. 12. It should be noted that in the case of chemical fixation of cultured neurons, shrinkage of dendrites and spines was negligible (Supplementary Fig. 14). This difference in the effect of shrinkage may be derived from the difference in the concentration of fixatives, the speed of chemical reaction, or the osmolarity of fixative solutions^{47, 48}.

In the supplementary information, the authors show that the spines do not change volume during the fixation process. Therefore, I am confused about the issue of chemical fixation and shrinkage from the data that is presented.

Again we thank the reviewer for asking the effect of chemical fixation in cultured neurons. As shown in Supplementary Figure 14, the morphology and size of spines are preserved after chemical fixation. This result indicates that shrinkage effect of fixatives may be different depending on whether they are applied to culture preparations or to tissue preparations. At present there is no theoretical framework that can systematically predict the extent of cell/tissue shrinkage after chemical fixation. We should mention that the shrinkage/expansion effects of chemical fixation depend on the concentration and type of fixatives, incubation time, osmolarity of the solution, and the speed of chemical reaction. In culture preparations, we selected the reduced concentration of glutaraldehyde (0.5% for fixation of cultured neurons vs 2.5% for fixation of the brain by cardiac perfusion). Fixation of cultured neurons can be performed in an environment of optimized osmolarity with rapid exposure to the chemicals. These experimental parameters are critical in preservation of cell structure and may help maintenance of spine volumes in culture preparations. We described this point in the Methods section as follows (underlined sentences):

“We compared our own spine volume data in cultured neurons expressing GFP (n = 1335) with the data of EM-reconstructed in vivo spines (n=938) available in the open data depository linked to the publication by Bloss et al²³. Both data were generated from hippocampal pyramidal neurons. The data for spine length in vivo were also taken from a

previous publication (n = 100)⁴⁶. Previous reports of direct comparison between chemical fixation and cryofixation reported that tissue shrinkage in the process of fixation, dehydration, and embedding for EM sample preparation was ~26% in the neuropil⁴⁷. We assumed that the shrinkage of spines should be proportional to the overall shrinkage of the neuropil. Based on this idea, we reduced the measured spine volumes in cultured neurons by the factor of 0.74 before generating the histogram shown in Supplementary Fig. 12. It should be noted that in the case of chemical fixation of cultured neurons, shrinkage of dendrites and spines was negligible (Supplementary Fig. 14). This difference in the effect of shrinkage may be derived from the difference in the concentration of fixatives, the speed of chemical reaction, or the osmolarity of fixative solutions^{47, 48}.

Also, the authors compare spine volumes from EM studies and from their own measurements and come up with a value of 26%. However, previous studies have not shown a shrinkage of spines in this order, only in the neuropil. I apologize if this is confusingly written here, but it's very confusing in the paper, and I am sure this can be addressed more succinctly.

We believe the previous arguments clarified our interpretation of the current results and the related previous literatures. Proportional shrinkage of spines, dendrites, and axons during chemical fixation is a straightforward interpretation and we utilized the shrinkage ratio of the whole neuropil taken from the previous literature as a reasonable estimate of the extent of spine shrinkage. We believe that the modified text in the Methods section will be useful to clarify our experimental design to the readers.

Reviewer #3 (Remarks to the Author):

Kashiwagi and colleagues have addressed all points adequately, that I had raised in my initial review. From my side I would favour publication of the manuscript in its current form.

We thank this reviewer for this positive response.

One small comment, nevertheless: The label in Figure 1 a 'Spine autodetection' means that a spine has the ability to detect itself, which I believe is not what was intended. 'Automated spine detection' appears more adequate.

We thank this reviewer for pointing this out. We corrected the label as suggested (Figure 1a).